# CRAG-MM: Multi-modal Multi-turn Comprehensive RAG Benchmark

## Abstract

Wearable devices such as smart glasses are transforming the way people interact with their surroundings, enabling real-time queries about entities in view. *Multi-Modal Retrieval-Augmented Generation (MM-RAG)* plays a key role in supporting such questions, yet no comprehensive benchmark exists for this task, particularly for wearable, *egocentric* scenarios. To fill this gap, we present **CRAG-MM**, a *Comprehensive RAG benchmark for Multi-modal Multi-turn* conversations. CRAG-MM contains a diverse set of 6.5K (image, question, answer) triplets and 2K visual-based multi-turn conversations across 13 domains, including 6.2K egocentric images designed to emulate wearable captures. We carefully constructed the questions to reflect real-world challenges, including five types of image-quality issues, six question types, varying entity popularity, differing information dynamism, and multi-turn dependencies. CRAG-MM defines three tasks: single-source augmentation, multi-source augmentation, and multi-turn QA—each paired with a dedicated retrieval corpus and *optional* APIs for image-KG and web search. Empirically, straightforward MM-RAG approaches achieve only 32% and 43% *truthfulness* on single- and multi-turn QA, respectively, while state-of-the-art industry solutions reach just 32% and 45%, revealing substantial headroom for future progress. The benchmark has hosted a leaderboard that attracted about a thousand participants, with winning solutions improving baseline performance by 28%, highlighting its early impact on advancing the field.

## 1 Introduction

*Wearable AI devices* are revolutionizing how people interact with computing systems. Modern wearable devices, such as Rayban Meta [1], Rabbit R1 [2], and the Humane AI Pin [3], enable vision-based conversations where users can ask questions about objects in their view. For example, a user may inquire about the history of a landmark they are viewing, the price of a product they are holding on different e-commerce websites, or repair instructions for a broken household device. What such questions often have in common is *the need for factual information that cannot be inferred from the images alone*, necessitating multi-modal Retrieval-Augmented Generation (MM-RAG) systems that access external sources for enriched and accurate responses. To advance the MM-RAG techniques, we construct the CRAG-MM benchmark, a comprehensive multi-modal benchmark of 6.5K single-turn and 2K multi-turn conversations, with an emphasis on use cases relevant to wearable AI devices. Specifically, our motivation is three-fold.

First, although many Visual Question Answering (VQA) benchmarks exist (Antol et al., 2015; Hudson & Manning, 2019; Schwenk et al., 2022), they rely primarily on common knowledge (Antol et al., 2015; Goyal et al., 2017; Schwenk et al., 2022) and visual reasoning (Hudson & Manning, 2019), which is insufficient to assess factual questions comprehensively. New benchmarks focusing on multi-modal knowledge integration have emerged in recent years. They nevertheless were constructed solely from Wikipedia (Lerner et al., 2022), created by templates (Talmor et al., 2021), or consisted of questions uncommon in real life (Talmor et al., 2021; Chang et al., 2022). CRAG-

---

[1] https://www.meta.com/ai-glasses
[2] https://www.rabbit.tech/rabbit-r1
[3] https://humane.com/

MM addresses these limitations by incorporating 8K conversations, with 89% focusing on factual questions that require external information for trustworthy answers.

Second, several question-answering (QA) benchmarks have emerged in recent years, covering *closed-book* (e.g., SimpleQA (Wei et al., 2024), FACTS Grounding (Jacovi et al., 2025)) and *open-book* (e.g., FreshQA (Vu et al., 2023)) QA. Some, like CRAG (Yang et al., 2024), specifically target the evaluation of RAG systems. CRAG-MM goes beyond them not only by introducing *multi-modal* use cases, but also by including 2k *multi-turn conversations*, where ∼38% of them involve domain shifts. This simulates natural topic drift, a common feature of human conversations, further enhancing the realism of the benchmark.

Third, existing VQA benchmarks typically feature high-quality images. In contrast, wearable devices often use wide-angle cameras, capturing *egocentric* images where objects of interest appear small, rotated, truncated, occluded, blurred, or poorly lit (Shenoy et al., 2024). To bridge this gap, the CRAG-MM benchmark includes 7.9K images, with 79% being egocentric, reflecting the real-world challenges in wearable AI applications.

As such, CRAG-MM is, to the best of our knowledge, *the first publicly released benchmark capable of effectively evaluating multi-modal RAG* and also *one of the earliest benchmarks designed to reflect wearable AI use cases*. The design of CRAG-MM follows the key features outlined in (Yang et al., 2024), with the following principles.

**Rich and insightful benchmark:** CRAG-MM is structured along four key dimensions—image quality, question type, entity popularity, and conversation complexity, mirroring real-world challenges faced by wearables QA systems and enabling in-depth analysis and debugging. It covers both common and challenging use cases; for example, 15% of images are low-quality egocentric images, 21% questions involve torso-to-tail entities; 52% questions are complex questions and require multi-source information synthesis, and 23% conversations contain multiple turns See Fig.1 and Fig.5 in Appendix A.1 for some examples from CRAG-MM.

**Reliable and fair evaluation:** To ensure fair comparison, we provide equal access to retrieval resources, including APIs for an image knowledge graph (KG) with 68K entries and a webpage corpus with 800K pages. The retrieval corpus is designed to mimic real-world conditions, with a 1:20 ratio of relevant to irrelevant information for image search and 1:2 for web search, allowing comparison of different solutions fairly and easily. The retrieval corpus, search APIs, and relevant metadata allow for evaluating different components of RAG systems: entity recognition, OCR, query rewrite, response generation, and so on.

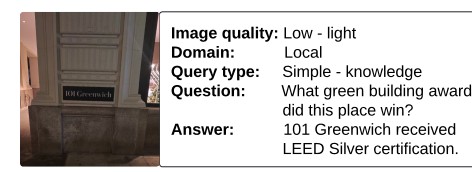

Figure 1: A CRAG-MM example.

**Challenging questions:** CRAG-MM contains realistic but hard questions. As shown in Tab.4, without RAG, the most advanced multi-modal large language model (MM-LLM) we evaluated (GPT-5 mini) achieved an accuracy of 37% on single-turn questions; straightforward solutions improved the accuracy to 50%; even the most advanced industry solutions (GPT-5), with access to potentially richer web corpus and better search engine, reached only an accuracy of 63% with 31% hallucinations. Multi-turn conversations contain simpler questions; still, the best industry solution achieved an accuracy of 70%, with 27% conversations being early stopped due to two consecutive erroneous or missing answers.

CRAG-MM has been used to host a leaderboard competition, attracting nearly 1K participants and 5K submissions.[4] The winning solution improved over straightforward solutions by 28%, highlighting its early impact on advancing the field.

**Differences from CRAG and other existing benchmarks:** To the best of our knowledge, CRAG-MM is the first comprehensive benchmark that focuses on wearable AI use cases. Different from CRAG (Yang et al., 2024) which targets text-based single-turn QA, CRAG-MM is a visual QA benchmark. Unlike other existing VQA benchmarks, CRAG-MM uniquely features wearable use

---

[4]Per the double-blind review policy, we withhold the competition name.

cases – questions are based on egocentric images and often require external knowledge to answer. It encompasses a variety of domains and question types, which can help reveal interesting insights and facilitate development. Further, CRAG-MM extends beyond single-turn QA by including multi-turn conversations, a common and essential use case for wearable devices. Tab.1 compares CRAG-MM to a few popular or recent benchmarks.

Table 1: Comparing CRAG-MM to existing benchmarks for factual question answering: SnapN-Tell (Qiu et al., 2024), WebQA (Chang et al., 2022), MultiModalQA (Talmor et al., 2021), MM-Vet (Yu et al., 2023), MT-Bench-101 (Bai et al., 2024), CRAG (Yang et al., 2024).

| Benchmark | Ego-centric | Image search API | Web search API | Multi-modal | Multi-turn | Torso & tail facts |
|---|---|---|---|---|---|---|
| SnapNTell | ✗ | ✗ | ✗ | ✓ | ✗ | ✓ |
| WebQA | ✗ | ✓[a] | ✓[b] | ✓ | ✗ | ✗ |
| MultiModalQA | ✗ | ✗ | ✗ | ✓ | ✗ | ✗ |
| MM-Vet | ✗ | ✗ | ✗ | ✓ | ✗ | ✗ |
| MT-Bench-101 | ✗ | ✗ | ✗ | ✗ | ✓ | ✗ |
| CRAG | ✗ | ✗ | ✓ | ✗ | ✗ | ✓ |
| CRAG-MM | ✓ | ✓ | ✓ | ✓ | ✓ | ✓ |

[a]Prefetched images.

[b]Prefetched snippets from wikipedia.

## 2 PROBLEM DESCRIPTION

An *MM-RAG QA system* takes as input an image $I$ and a question $Q$, and outputs an answer $A$; the answer is generated by MM-LLMs based on information retrieved from external sources, combined with knowledge internalized in the model. A *Multi-turn MM-RAG QA system* in addition takes questions and answers from previous turns as context to answer new questions. The answer should provide useful information to answer the question, without adding any hallucination.

### 2.1 QUESTION TYPES

We first define six types of questions as follows.

- *Simple-recognition*: Questions asking for simple facts that can be directly answered from the image, e.g., *"what brand is the milk"* or *"who wrote this book"*, where the brand name and the book author are shown on the image.

- *Simple-knowledge*: Questions asking for simple facts that require external knowledge to answer, e.g., *"what's the price of this sofa on Amazon"*.

- *Multi-hop questions*: Questions that require chaining multiple pieces of information to compose an answer, such as "what other movies has the director of this movie directed".

- *Comparison questions*: Questions require comparing multiple pieces of information, such as *"is this cheaper on Amazon"* (where the image is showing a product and its store price).

- *Aggregation questions*: Questions require aggregating multiple pieces of information, *"which drinks do not contain added sugar among these"* (where the image is showing a few drinks in a grocery store).

- *Reasoning questions*: Questions about an entity that cannot be directly looked up from the retrieved content and require reasoning to answer, such as *"can the dryer be used in Europe"* (where the image shows a dryer).

### 2.2 TASKS

We provide a unified retrieval corpus to ensure fair comparison across methods. To support rapid prototyping, we additionally offer *optional* mock search APIs; users are not required to use them and may instead build stronger custom retrievers using the retrieval corpus. Based on the available

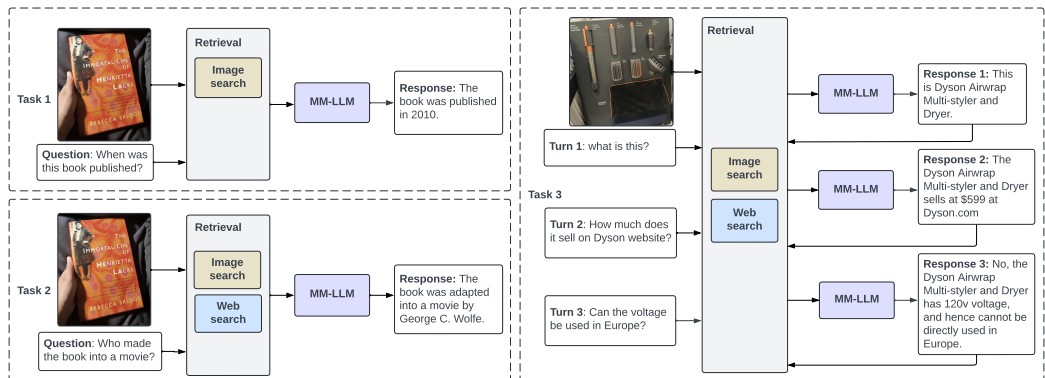

Figure 2: CRAG-MM task design.

retrieval sources, we define three evaluation tasks. As shown in Fig.2, Task 1 and Task 2 contain single-turn questions, where the former provides image KG for retrieval, and the latter additionally provides web sources; Task 3 focuses on multi-turn conversations. The three tasks can guide users of the benchmark in building end-to-end MM-RAG systems.

**Task 1: Single-source augmentation.** Task 1 tests basic MM-RAG capabilities using an image-based *mock KG*. The mock KG is indexed by the image, and stores structured data associated with the image; answers to the questions may or may not exist in the mock KG. We additionally provide an optional image search *mock API* that takes an image as input, and returns visually similar images and their metadata to support answer generation.

**Task 2: Multi-source augmentation.** Task 2 tests how well MM-RAG systems synthesize information from different sources. In addition to image KG, we provide a webpage corpus containing both relevant information and realistic noise. Similarly, we provide an optional web search mock API to search the webpage corpus.

**Task 3: Multi-turn QA.** Task 3 tests context understanding in multi-turn conversations. Each conversation contains 2–6 turns. Questions in second turn or beyond may or may not need the image for answering the questions.

## 3 DATA DESCRIPTION

We now describe the benchmark dataset. CRAG-MM covers 13 domains and six types of questions, all in English. It contains three parts of data: the image set, the QA set, and the contents for retrieval.

### 3.1 IMAGE SET

CRAG-MM contains two types of images: *egocentric* and *normal*. The egocentric images were collected using smart glass devices from first-person perspective; the normal images were collected from publicly available images. We collected both types of images via two methods: *pre-defined collection* and *free-form collection*.

For pre-defined collection, we first specified a list of domains and entity types likely to be involved in user wearable interactions, such as *electronic products* in the *shopping* domain. We adopted the method described in Sun et al. (2023) and sampled entities of top, middle and bottom popularity. We defined the popularity based on some publicly available traffic information for each entity type, and created equal number of questions for each bucket. We recruited vendors to collect images for the sampled entities using the RayBan Meta Smart Glasses[5] whenever possible; otherwise, we searched online to find an image for the target entity from public sources.

---
[5]https://www.meta.com/ai-glasses

For free-form collection, we asked the vendors to wear a pair of smart glasses and interact in their daily lives. We recommended to collect $80\%$ images from the provided domains, and $20\%$ for any additional use cases that the vendors found plausible for interaction with smart glasses.

We collected 6.2K egocentric images and 1.7K public images in total. We also requested $15\%$ of the egocentric images to be captured under imperfect conditions: low-light, blurred, truncation, occlusion, or rotation. See Tab.6 in Appendix A.1.1 for the definition and Tab.2 for the distribution.

### 3.2 Question Answering data

**Single-turn QA data:** We constructed the QA set for the collected images based on two sources—KGs or web contents. For the QA set constructed from KGs, we first leveraged existing entity types and relations in the KGs to create meaningful QA templates. We then used entities sampled from the image collection to pair with the question templates and took the associated attribute values as the answer. For the QA set constructed from web contents, we asked annotators to create plausible questions for wearable devices that could possibly be answered by web search. The annotators then also created the complete web search query based on the image and the question, and recorded the ground truth answers. See Appendix A.2 for more details.

Table 2: Distribution by image quality.

| Type | Egocentric | Normal | Total |
|------|-----------|--------|-------|
| Normal | 5143 | 1593 | 6736 |
| Low-light | 267 | 33 | 300 |
| Blurred | 191 | 24 | 215 |
| Truncated | 370 | 31 | 401 |
| Occluded | 116 | 8 | 124 |
| Rotated | 161 | 6 | 167 |
| Total | 6248 | 1695 | 7943 |

**Multi-turn QA data:** We first created a number of simple questions based on the web contents as first-turn seed questions. We then prompted `Llama-3.2-90B-Vision-Instruct` (Dubey et al., 2024; Meta AI, 2024) to create multi-turn conversations based on the given seed questions. Each conversation has two to six turns, and covers one to three domains. Next, we asked the annotators to review the conversation, remove or revise turns that are not plausible, too simple, or do not have a single indisputable answer. Finally, the annotators also created the ground truth answers for each turn by conducting web search.

Table 3: Distribution of domains and question types for CRAG-MM single- and multi-turn QA. We use the first turn's domain for multi-turn conversation.

| | Simple-rec. | Simple-know. | Multi-hop | Comp. | Agg. | Reason. | Single-turn | Multi-turn |
|------|------|------|------|------|------|------|------|------|
| Animal | 10 | 191 | 67 | 115 | 71 | 66 | 520 | 148 |
| Book | 12 | 53 | 37 | 17 | 12 | 9 | 140 | 147 |
| Food | 35 | 312 | 108 | 132 | 99 | 85 | 771 | 196 |
| General Obj. Rec. | 26 | 210 | 82 | 74 | 43 | 50 | 485 | 184 |
| Local | 29 | 428 | 168 | 119 | 106 | 87 | 937 | 181 |
| Math & Science | 61 | 62 | 16 | 17 | 18 | 20 | 194 | 130 |
| Plants & Gardening | 19 | 443 | 119 | 125 | 117 | 107 | 930 | 118 |
| Shopping | 52 | 239 | 48 | 66 | 41 | 48 | 494 | 204 |
| Sports & Games | 12 | 131 | 31 | 24 | 21 | 12 | 231 | 123 |
| Style & Fashion | 13 | 57 | 15 | 14 | 5 | 9 | 113 | 115 |
| Text Understanding | 170 | 113 | 15 | 10 | 21 | 18 | 347 | 113 |
| Vehicle | 14 | 250 | 147 | 196 | 113 | 140 | 860 | 120 |
| Other | 9 | 125 | 75 | 68 | 105 | 58 | 440 | 177 |
| Total | 462 | 2614 | 928 | 977 | 772 | 709 | 6462 | 1956 |

We collected 6.5K single-turn questions and 2K multi-turn conversations with an average length of 4.9 turns in the final dataset. Tab.3 summarizes distribution of the questions across different dimensions. The size of each slice allows us to get metrics with $<5\%$ margin-of-error (with $95\%$ confidence level) for most of the slices.

## 3.3 RETRIEVAL CONTENTS

To ensure consistent accessibility and fair comparison across methods, we provide all retrieval contents as part of CRAG-MM. These contents span two complementary sources: an image-based corpus and a text-based web corpus, designed to reflect the challenges faced by real-world MM-RAG systems. Both corpora capture naturally occurring noise and ambiguity present in wearable QA.

### 3.3.1 RETRIEVAL CORPUS

**Image-based corpus.** We curate an image–metadata corpus: each entry contains an image and its associated structured attributes (e.g., plant name and species). The corpus consists of 68K images covering 26K entities and captures 93% of all entities referenced in CRAG-MM questions (Fig. 3).

**Text-based corpus.** We further include a web-derived text corpus. Webpages are chunked into 512-token segments and embedded using BGE (Xiao et al., 2024), yielding 2.7M chunks spanning 800K urls. The resulting corpus achieves an estimated 89% recall for CRAG-MM questions (Fig. 3), offering broad but imperfect coverage typical of real-world web search.

### 3.3.2 MOCK SEARCH APIS

CRAG-MM additionally provides *optional* search APIs that mirror retrieval interfaces used in practical MM-RAG pipelines. These APIs enable rapid development and reproducible evaluation, especially for leaderboard participants. They are nevertheless not required – users may instead build custom retrievers directly on the provided corpora.

**Image-based KG search API.** Given an input image, this API retrieves visually similar images and associated metadata from the image-based corpus using CLIP ViT-L/14@336px (Radford et al., 2021) embeddings. The API exposes the real-world retrieval challenges described above: low lighting, occlusion, and long-tail entities. As a result, directly querying the index using raw query images yields only 52% recall (Fig. 3), illustrating the intrinsic difficulty of visual retrieval under realistic egocentric conditions.

**Text-based web search API.** Given a text query, this API retrieves top-ranked webpages from the web corpus. At inference time, the query is embedded and matched against chunk embeddings to return the most relevant documents. See Appendix A.3.3 for additional construction details.

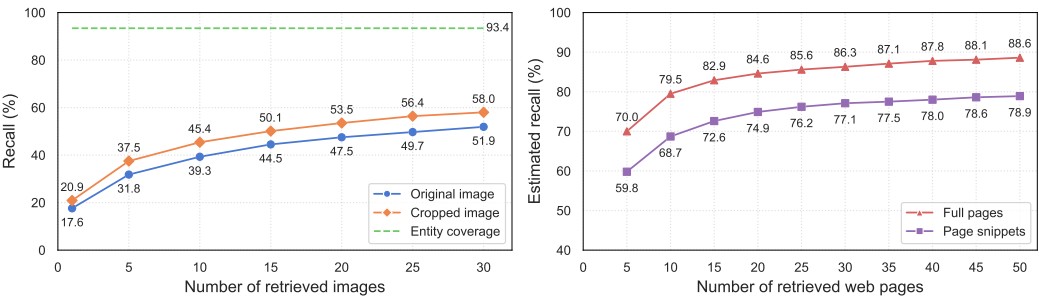

Figure 3: **Image search recall (left)**: the image search index covers 93.4% of the query entities (oracle recall). However, using the full image to query the index only achieves 51.9% recall. Manual cropping improves the recall slightly to 58.0%. **Web search recall (right)**: the top 50 web search results are estimated to contain the ground truth facts for 88.6% questions.

### 3.3.3 HOW TO USE THE RETRIEVAL CONTENTS OF CRAG-MM?

CRAG-MM supports multiple evaluation modes tailored to different research goals. Given the question–answer data, retrieval corpus, and search APIs, the benchmark enables controlled evaluation of retrieval, generation, or the full MM-RAG pipeline.

**Mode 1: API-Based Retrieval & Summarization.** Users can directly invoke CRAG-MM's search APIs to rapidly prototype MM-RAG systems. This mode is particularly useful for challenge partici-

pants who need a ready-to-use retrieval interface. Developers can focus on query formulation, image cropping and preprocessing, information extraction, and grounded answer generation, without the overhead of building custom indices or retrievers. We use this mode for evaluating the straightforward and leaderboard winning solutions in Section 5.2.1

**Mode 2: Custom Retrieval & Summarization** Users can instead build and evaluate their own retrievers using CRAG-MM's shared retrieval corpus. This mode supports research on end-to-end RAG pipelines with controlled retrieval inputs. The equally accessible corpus ensures fair comparison across methods and eliminates confounds stemming from unequal access to proprietary search engines or non-replicable web indices.

**Mode 3: Black-Box End-to-End Evaluation.** CRAG-MM can evaluate MM-RAG systems whose retrieval modules are not exposed to the user. In this setting, the benchmark assesses the entire retrieval–generation stack in a black-box manner. This is the mode we use to benchmark commercial state-of-the-art MM-RAG systems in Section 5.2.2.

### 3.3.4 DATA SPLIT

We split the data in two steps. We first held out two domains for private test only, for testing the generalization of the MM-RAG systems. We then split the remaining domains' data into three sets with similar distributions: *validation, public test*, and *private test*, containing 30%, 30%, 40% of the data respectively. We released the validation and public test sets on HuggingFace.[6]

## 4 METRICS AND EVALUATION

We adopt the metrics and evaluation methods from Yang et al. (2024) and extend them to the multi-turn setting.

**Single-turn QA.** We score each model response as *correct*, *missing* (e.g., "I don't know," "I can't find ..."), or *hallucinated* (incorrect or irrelevant), assigning scores of $1$, $0$, and $-1$ respectively. We report **Truthfulness** as the average score.

**Multi-turn QA:** Since here is no universally accepted evaluation strategy for multi-turn interactions, we adapted the approach of Bai et al. (2024), which aligns most closely with information-seeking tasks. Specifically, we *early-stop* a conversation once the model produces two consecutive missing or hallucinated answers, and treat all remaining turns as missing, mirroring realistic user behavior when experiencing repeated failures. We then compute **Truthfulness** as the average score across all turns in a conversation, and report the grand average across all conversations.

**Auto-evaluation:** We adopt LLM-as-a-judge to evaluate answer quality. The judge attains 99.1% accuracy against manual annotations (Tab.7). See Appendix A.4 for additional details.

## 5 BENCHMARKING

In this section, we evaluate several visual QA systems on the CRAG-MM test set and demonstrate how the benchmark reveals challenges and points to directions for advancing MM-RAG research. We address the following three research questions (RQ) through our experiments.

**RQ1:** Does CRAG-MM reveal new challenges beyond straightforward solutions?

**RQ2:** How well do state-of-the-art industry solutions perform on CRAG-MM?

**RQ3:** What quality improvement directions does CRAG-MM suggest?

### 5.1 EXPERIMENT DESIGN

**Straightforward solutions.** We evaluate both MM-LLM-only baselines and three straightforward RAG variants for three representative models: `Llama-3.2-90B-Vision-Instruct`, `Gemini-2.5-Flash` (Comanici et al., 2025), and `GPT-5-mini` (OpenAI, 2025). This setting

---

[6]Per the double-blind review policy, we withhold the url for the released data.

Table 4: Performance of straightforward solutions on CRAG-MM single- and multi-turn QA. All numbers are percentages. **Bold** indicates best quality and *Italic* indicates best within the group. Even the best MM-LLM achieves only 18% truthfulness for single-turn QA and 30% for multi-turn QA; straightforward MM-RAG solutions improve only to 32% for single-turn and 43% for multi-turn.

| | | Model | Acc. | Miss. | Hallu. | Truth. | Early Stop. |
|---|---|---|---|---|---|---|---|
| **Single-turn** | **MM-LLM** | Llama 3.2 90B | 28.2 | 39.1 | 32.8 | -4.6 | - |
| | | Gemini 2.5 Flash | 36.6 | *38.0* | 25.4 | 11.2 | - |
| | | GPT-5 Mini | *37.4* | 43.7 | *19.0* | *18.4* | - |
| | **Task 1** | Llama 3.2 90B | 13.5 | 71.0 | **15.6** | -2.1 | - |
| | | Gemini 2.5 Flash | 37.9 | *36.8* | 25.3 | 12.6 | - |
| | | GPT-5 Mini | *39.3* | 43.9 | 16.8 | *22.5* | - |
| | **Task 2** | Llama 3.2 90B | 30.1 | 50.1 | 19.8 | 10.3 | - |
| | | Gemini 2.5 Flash | **49.9** | *22.6* | 27.5 | 22.4 | - |
| | | GPT-5 Mini | 48.7 | 34.1 | *17.2* | **31.5** | - |
| **Multi-turn** | **MM-LLM** | Llama 3.2 90B | 42.2 | *25.0* | 32.8 | 12.7 | 64.7 |
| | | Gemini 2.5 Flash | 29.2 | 57.5 | **13.4** | 16.5 | 88.1 |
| | | GPT-5 Mini | *48.9* | 34.0 | 17.1 | *30.4* | *60.8* |
| | **Task 3** | Llama 3.2 90B | 37.1 | 46.1 | 16.9 | 18.9 | 81.7 |
| | | Gemini 2.5 Flash | 54.4 | 24.2 | 21.4 | 31.4 | 55.8 |
| | | GPT-5 Mini | **61.0** | **22.5** | *16.5* | **42.5** | **43.5** |

uses fixed retrieval inputs produced by the CRAG-MM search APIs (Mode 1; Section 3.3.3) and measures each model's ability to generate grounded answers from real-world, noisy retrieval content. Implementation details are summarized below, with full descriptions in Appendix A.5.1.

- **MM-LLM-only:** A simple prompting strategy encouraging concise answers and explicit abstention *"I don't know"* when uncertain.
- **Task 1:** Query the image search API using the entire image; retrieve the top 30 results and filter by a similarity threshold of 0.75. Retrieved metadata are concatenated into a 2K token context window.
- **Task 2:** Apply query rewriting for web search; concatenate webpage snippets as reference text, expanding the context window to 8K tokens (similar to Kandpal et al. (2023) and Mallen et al. (2023)).
- **Task 3:** Combine multi-source retrieval with conversation history for context.

**Leaderboard winning solutions.** The leaderboard winning systems on CRAG-MM [7] also operate in Mode 1 (Section 3.3.3). Unlike the straightforward baselines, these systems can optimize image cropping and pre-processing, query formulation and apply custom filtering or re-ranking of retrieved results rather than relying on fixed retrieval inputs.

**Industry state-of-the-art solutions.** We further benchmark commercially deployed MM-RAG systems that couple proprietary MM-LLMs with search engines. We query each system with CRAG-MM questions and evaluate its outputs in a fully black-box manner (Mode 3; Section 3.3.3). See Appendix A.7 for additional implementation details.

## 5.2 RESULT

### 5.2.1 RQ1: NEW CHALLENGES BEYOND STRAIGHTFORWARD MM-RAG SOLUTIONS

Tab.4 highlights several observations. First, the best MM-LLM-only model (GPT-5 Mini) reaches only 37% accuracy and 18% truthfulness on single-turn QA, and 49% accuracy and 30% truthfulness on multi-turn QA, showing CRAG-MM is challenging without retrieval. Next, for single-turn QA, vanilla image-KG retrieval (Task 1) yields only a modest improvement (+4% Truthfulness). Adding web search (Task 2) improves accuracy but also increases hallucination rates; the highest accuracy

---

[7] We withhold citations to winning team reports for double-blind policy.

and truthfulness remain only 50% and 32%, respectively. Finally, multi-turn QA follows the same trend, revealing a persistent gap even after adding straightforward retrieval. See Appendix A.6 for an extensive study including more models. Further, to examine whether the low image search recall alone drives the difficulty, we re-evaluated models by excluding the subset of questions where entity was not retrieved by the image search API. As shown in Appendix A.6.2, accuracy remains only 56% and 61% on single- and multi-turn, and truthfulness only 39% and 42%, highlighting that the task remains challenging beyond image retrieval.

### 5.2.2 RQ2: INDUSTRY SOTA AND LEADERBOARD WINNING SOLUTIONS

We report results from two groups of solutions: leaderboard winning and industry SOTA solutions. The leaderboard solutions typically combined multi-task learning with supervised fine-tuning on `Llama-3.2-11B-Vision-Instruct` model (Meta AI, 2024), and further improved quality through knowledge distillation from larger models (e.g., `Llama-3.2-90B-Vision-Instruct` (Meta AI, 2024)). For comparison, we also include the straightforward MM-RAG solution built on the same model.

Tab.5 summarizes the results. Note that truthfulness scores are not perfectly comparable across all groups: leaderboard and straightforward solutions rely on the CRAG-MM search APIs, whereas industry systems typically leverage substantially stronger proprietary search engines and indices; leaderboard scores are reported on the private test set, whereas other systems use the public test set. Nevertheless, the performance trends remain consistent. We highlight two key observations.

First, industry SOTA systems achieve accuracy improvements over straightforward MM-RAG (+13% single-turn; +9% multi-turn), yet their truthfulness remains similar (+0% single-turn; +2% multi-turn). Hallucination rates remain high: 31–49% for single-turn and 26–35% for multi-turn, indicating that even advanced commercial systems struggle to produce reliable answers in realistic egocentric scenarios. Second, leaderboard winning solutions, though not as strong as industry systems overall, achieve substantial improvements over straightforward MM-RAG using the same base model (+28% single-turn; +18% multi-turn). They also achieve the lowest hallucination rates across all systems, largely due to more conservative and calibrated abstention behavior. This comes at the cost of lower accuracy and higher missing rates, suggesting a clear direction for improving calibrated confidence and retrieval quality.

Table 5: Performance of leaderboard winning and industry SOTA systems on CRAG-MM.

|  | System | Acc. | Miss. | Hallu. | Truth. | Early Stop. |
|---|---|---|---|---|---|---|
| **Single-turn** | Llama 3.2 11B RAG | *35.3* | *20.8* | 43.9 | -8.6 | - |
|  | Winning team | 29.3 | 61.2 | **9.6** | *19.7* | - |
|  | Claude Sonnet | 45.7 | 5.8 | 48.5 | -2.8 | - |
|  | Gemini | 58.2 | *3.1* | 38.7 | 19.5 | - |
|  | GPT-5 | **62.7** | 6.8 | *30.5* | **32.2** | - |
|  | SG MMAI | 39.6 | 7.4 | 52.9 | -13.3 | - |
| **Multi-turn** | Llama 3.2 11B RAG | *46.6* | *11.0* | 42.4 | 8.8 | *63.7* |
|  | Winning team | 44.4 | 40.2 | **15.4** | *26.6* | 65.9 |
|  | Claude Sonnet | 58.7 | 6.6 | 34.6 | 28.6 | **25.0** |
|  | Gemini | 66.2 | *3.5* | 30.3 | 30.1 | 37.0 |
|  | GPT-5 | **70.0** | 3.9 | *26.1* | **45.0** | 26.9 |
|  | SG MMAI | 42.3 | 7.6 | 50.1 | 1.0 | 67.6 |

### 5.2.3 RQ3. DIRECTIONS FOR QUALITY IMPROVEMENTS

We further analyze the leaderboard and industry SOTA systems on fine-grained slices of CRAG-MM in Fig.4. (See Fig.8 in AppendixA.6.1 for results on straightforward solutions.)

First, *all systems show substantial degradation under challenging visual conditions*, with truthfulness dropping by up to 46% on low-quality images ( Fig.4a). The hardest categories are "Low-light" and "occluded", where the best systems reach only 20% and 24%. Notably, leaderboard solutions show far smaller variance across visual conditions, indicating that robustness gains are possible even

without stronger retrievers. Second, *entity recognition is considerably harder when relying solely on visual signals than when text cues are available* (up to 37% drop; Fig.4b). Performance further declines for less popular entities (Fig.4c), reflecting both visual ambiguity and the sparsity of information for long-tail entities – an important challenge in wearable scenarios. Third, *systems struggle with questions requiring specific factual details (Simple-knowledge), multi-evidence composition (comparison and aggregation), or multi-hop retrieval* (Fig.4d), highlighting the need for improved retrieval recall, multi-modal grounding, and reasoning over heterogeneous evidence. Importantly, industry SOTA systems use proprietary search engines and indexing rather than the CRAG-MM search APIs; their low accuracy and truthfulness therefore underscore the intrinsic difficulty of ego-centric, wearable QA. Finally, *maintaining coherent multi-turn dialogue remains a grand challenge*. Consistent with findings in Laban et al. (2025), systems show significant quality drops (up to 22% drop) when the current question depends on prior conversational context (Fig.4e). Even the best system results in 27% of conversations being early-stopped (Tab.5), with only 3.2 successful turns out of 4.9 on average (Fig.4f).

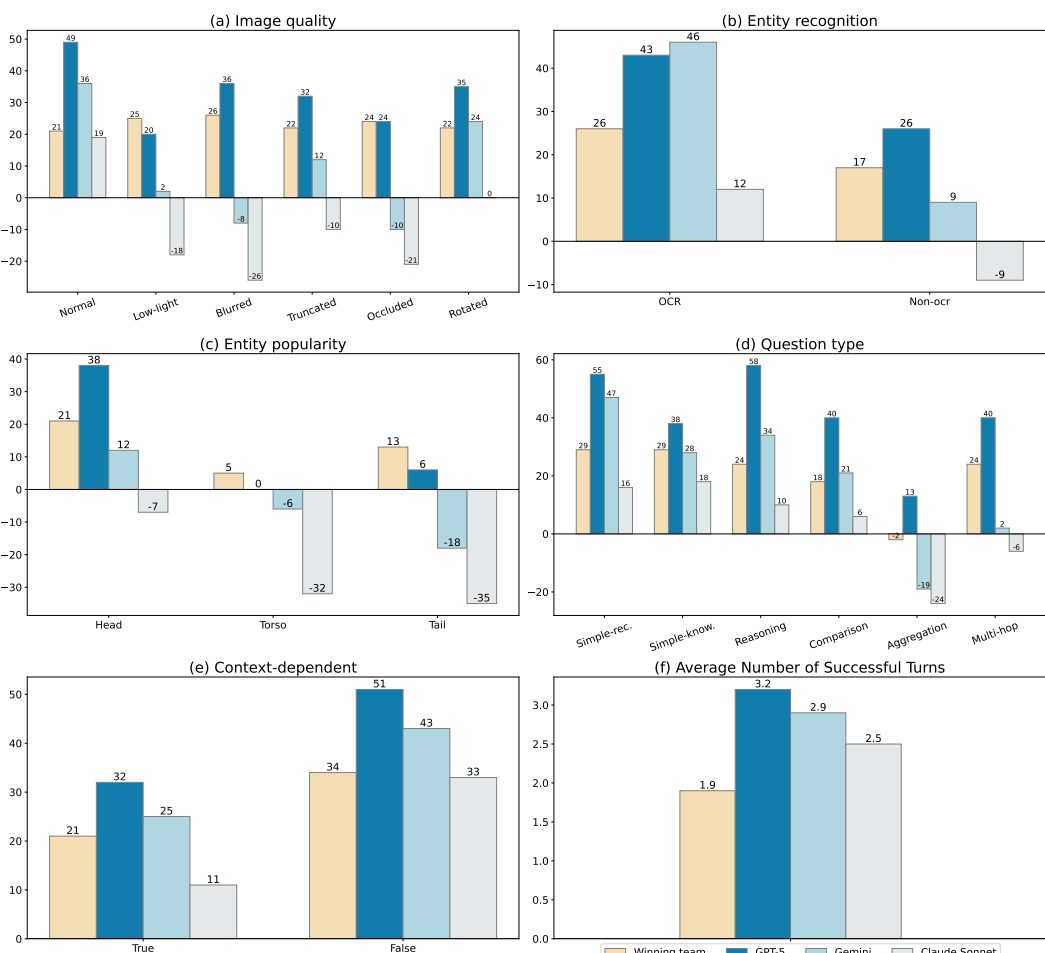

Figure 4: Winning and SOTA systems' performance across different dimensions. Figure (a)–(e) show truthfulness; (f) shows the average number of successful turns in multi-turn QA.

## 6    CONCLUSION

In this paper, we propose CRAG-MM, the first MM-RAG benchmark tailored to wearable AI applications. Its principled design and accessible search APIs empower systematic evaluation of MM-RAG capabilities and inform directions for future advances.

## REPRODUCIBILITY STATEMENT

We made the following efforts to make sure our reported results are reproducible:

1. We describe the detailed experiment setup to benchmark the MM-LLM-only, straightforward MM-RAG, and industry SOTA solutions in Appendix A.5 and Appendix A.7. Specifically, we include the implementation details and the prompts used for getting responses from the MM-LLM-only and straightforward solutions for each task. We also include the API parameters (models and tool usage) used for evaluating the industry SOTA systems.

2. We disclose the LLM-judge and the prompt used in the evaluation in Appendix A.4.

3. We use Github repositories to host the code for search API and evaluation experiments, but withhold disclosing the links per the double-blind review policy.

4. In addition, we provide details of how the benchmark dataset was created in Section 3 and Appendix A.1, including the process of image collection, question answering data creation, and search API construction.

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

# A APPENDIX

## A.1 DATASET

### A.1.1 DEFINITION OF IMAGE QUALITY

Table 6: Definition of low quality images.

| Category | Definition |
|---|---|
| Low-light | The entity referred in the query is illuminated by a small amount of light, such as at dusk, during night time, or in shadowed areas. |
| Blurred | The entity referred in the question appears to be fuzzy, smeared, or indistinct in the image. |
| Truncated | The entity referred in the query is partially out of the image. |
| Occluded | There is something, like a finger, in the way of the camera and the entity. E.g. the entity is behind a chain link fence or a window screen. |
| Rotated | The entity referred in the query is more than 10 degrees off from a right orientation. |

### A.1.2 EXAMPLES OF CRAG-MM QA

Fig.5 provides some single-turn QA examples from CRAG-MM.

## A.2 QA DATA CREATION

All single-turn (Tasks 1 and 2) and multi-turn (Task 3) QAs were created using a human-in-the-loop pipeline. The design dimensions for CRAG-MM, domains, image-quality issues, question types, were inspired from real user interactions with a commercial wearable device, ensuring the dataset reflects authentic wearable-AI usage patterns. Trained annotators followed detailed guidelines to create plausible, user-initiated queries; and performed multi-stage reviews to filter out trivial, unnatural, or template-like questions. We detail the annotation workflow and quality assurance measures below.

### A.2.1 ANNOTATION PROCESS AND QUALITY CONTROL

We designed a rigorous process for annotation quality assurance, including 5 stages: guideline design, annotator qualification, QA and metadata curation, verification and correction, and expert audit.

Stage 1: Guideline Design. Annotators were tasked with identifying fine-grained entities (e.g., "Golden Retriever" rather than "Dog") within images and generating question-answer pairs and associated metadata such as image-query category (e.g. plant, local) and question type. We enforced strict constraints to ensure validity. These constraints are designed to comply with our benchmark principles and mitigate common bias in VQA dataset.

- Visual grounding: to prevent models from relying on language priors Goyal et al. (2017); Agrawal et al. (2018), annotators were required to use anaphoric pronouns (e.g., "Where is this building"), rendering the query textually ambiguous and forcing the model to rely on visual context to resolve the reference.
- Stratified query type distribution: to prevent the benchmark from skewing towards either trivial recognition or purely textual trivia, we enforced a ∼50/50 split between simple and complex queries defined in 2.1. This addresses answer-distribution bias discussed in Marino et al. (2019).
- Factual determinism: questions were restricted to those with indisputable, deterministic answers. Subjective (e.g., "Is this cute") or open-ended questions were prohibited to ensure consistent evaluation.
- Safety & privacy: Face blur was properly applied prior to annotation. Images containing PII or integrity violations will be rejected.

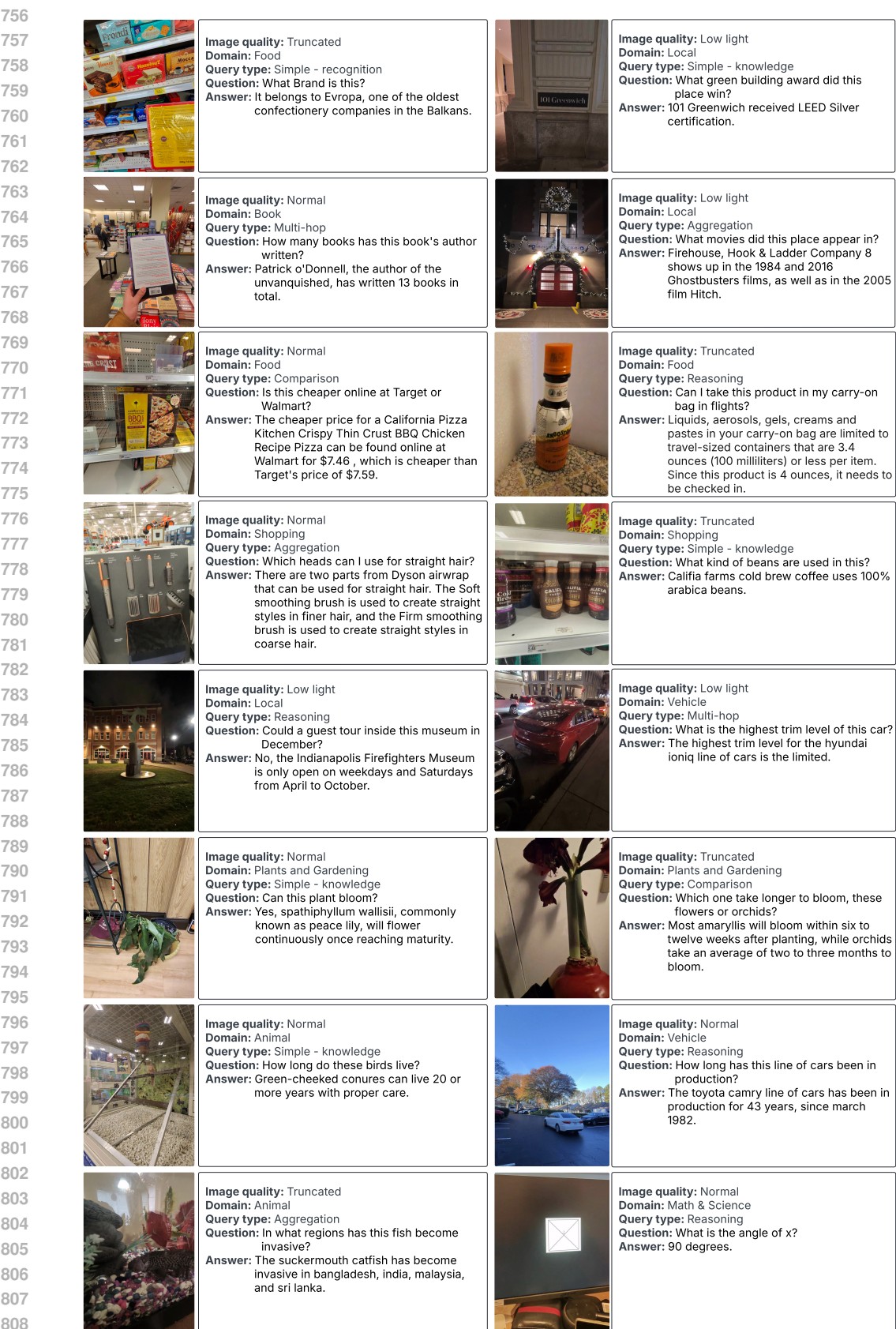

Figure 5: CRAG-MM examples.

- Other constraints: such as to avoid repeating queries, avoid asking questions that's too general such as "what is this", and avoid asking questions with yes/no answers.

Stage 2: Annotator Qualification. Candidates need to review the guidelines, pass a knowledge quiz, and complete a training queue with 100 QA creation jobs. Qualification required passing a high-quality threshold evaluated against a fine-grained error rubric.

Stage 3: QA and Metadata Curation. Qualified annotators proceeded to the production phase to construct image-question-answer triples. For each image, annotators were tasked with: (1) identifying valid fine-grained entities; (2) formulating a visual-anaphoric question, e.g. "when was this building built"; and (3) providing a deterministic answer backed by a verified source URL and associated metadata (e.g., entity source, query dynamism).

Stage 4: Verification and Correction. Following initial curation, a separate pool of high-quality annotators conducted a full-pass verification, mitigating self-confirmation bias—where annotators overlook ambiguities in their own writing. Annotators focused on correcting metadata errors, fixing grammatical issues and rejecting samples with data validity or safety issues. This stage resulted in:

- ∼25% of single-turn QAs modified and 2% of QAs rejected due to irresolvable quality or safety violations.
- 79% of multi-turn conversations with at least one human-edited query or answer, with an average of 4.8 turns modified per conversation.

Stage 5: Expert Audit. A final quality audit was conducted by three in-house experts on a random subset. They assessed the quality of the triples and the correctness of the associated metadata, achieving 90% inter-annotator agreement rate.

### A.2.2 QA SET CONSTRUCTED FROM KGS

Starting from the entities sampled during the image collection, we created a set of simple-knowledge QA data following three steps.

1. For each entity type, we selected a meaningful relation $(e, r)$ and created question templates. For example, for *(product, brand)*, we create a question *"what is the brand of this product?"*.

2. We then used entities sampled for the image collection to pair up with the question. We created equal number of questions for head, torso and tail entities.

3. Last, we took the associated attribute values as the answer to the question to obtain question answer pairs.

We created the *Comparison, Aggregation, and Reasoning* questions in a similar way but made sure the sampled subject entities can form meaningful questions given the question type. We used heuristics to select entity types for such questions.

Finally, we created multi-hop questions in 3 steps, similar to Talmor & Berant (2018). We first sampled an entity $e_1$ from the KG, and selected 2 relation triplets following a two-hop path: $(e_1, r_1, e_2)$ and $(e_2, r_2, e_3)$. We then created a question template describing the path. For example, for path *(book$_1$, author, person)* followed by *(author, write, book)*, we created the template *"what is the latest book of this book's author?"*. The answer to the question will be contained in $e_3$ in the second triplet.

### A.2.3 QA PAIRS CONSTRUCTED FROM WEB CONTENTS

We created QA pairs from web search results, in three steps.

1. Given the collected images, we asked annotators to write down questions that are plausible to interact with wearable devices and could possibly be answered by web search. For example, *"how much is this vehicle?"* (where the image is showing a 2024 volkswagon tiguan).

2. Next, the annotators will create web search queries based on the image and the question, such as *"how much is the 2024 volkswagon tiguan?"* for the above example. Then they will search the web using the search query to find relevant result for answering the question. Note that we only do text-based web search and rely on image search to provide the entity name.

3. Finally, the annotators note down the ground truth answers based on the web search results.

### A.2.4 MULTI-TURN QA CONSTRUCTION

While LLMs were used to draft candidate questions and answers for multi-turn conversations, human annotators validated and revised every conversation. For each sample, annotators ensured (1) the naturalness of the query, (2) appropriate difficulty, and (3) factual correctness of the answer. They removed or rewrote queries that are too simple or unnatural, corrected factual errors in the answers, and annotated all associated metadata. Our preliminary analysis showed that this hybrid approach yields greater question diversity than fully human-crafted data alone.

### A.2.5 PRIVACY AND CONSENT PROTOCOLS

We took multiple steps to ensure that privacy and consent requirements were strictly followed.

All images were collected by vendors who signed an agreement. The agreement explicitly states "You will only use the Products within your home and approved outdoor locations and will not allow any other person to be recorded by the Products unless such person is a household member who is at least 18 years of age and has signed a separate household member release ("Household Member Release")." We also explicitly instructed the vendors not to capture people or privacy-sensitive information (e.g., IDs, phone numbers, emails, etc).

In addition, we followed a rigorous and extensive process to obtain the privacy approval for data release. As part of this process, all identifiable faces were blurred to ensure privacy.

Finally, we have carefully reviewed each image and removed the ones with any sensitive information or safety violation. This includes privacy-sensitive contents or topics related to child abuse, dangerous organizations, hate speech, mental health, and politics.

## A.3 SEARCH API CREATION

### A.3.1 IMAGE SEARCH API CREATION

We constructed the image-based mock KG in four steps.

1. We first included the images used to generate the questions in Section A.2.2, and defined coverage categories based on these seed images.

2. Second, we collected candidate data from the open web (e.g., Wikipedia), including fine-grained entities within each category, their metadata, images, and inter-entity relationships.

3. From this pool, we selected candidate images to populate the mock KG. To ensure entity coverage, we added positive examples. To create a more realistic testing environment, we also added hard negatives, i.e. images visually similar to the query images but associated with different entities. Formally, let $\mathbf{I}$ denote the set of images for the entity set $\mathbf{E}$ that we used to construct the KG question answer data. For each $i \in \mathbf{I}$, we added up to 30 hard negatives, denoted by $S(i)$. We further augmented the set with randomly sampled images for the entities in $S(i)$, denoted as $E(S(i))$, i.e., $\mathbf{E} \cup \bigcup_{i \in \mathbf{I}} E(S(i))$.

4. Finally, we removed the query images, simulating a realistic setting in which user queries are absent from the KG, but visually similar or related images remain.

All images in the mock KG were encoded with `CLIP ViT-L/14@336px`, and we built the image retrieval index using ChromaDB(Chroma).

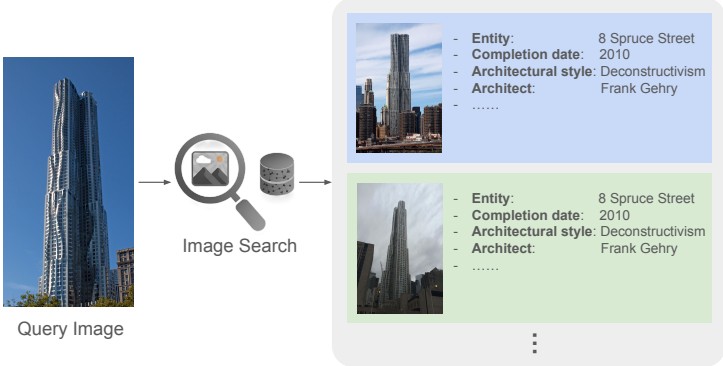

Figure 6: An example of the image search API.

### A.3.2 IMAGE SEARCH RECALL

Fig.7 shows the recall of the image search API for the two image types by using a crude method (query by the original image). Egocentric images has lower recall compared to the normal images, showing the difficulty of entity recognition and image understanding on egocentric images.

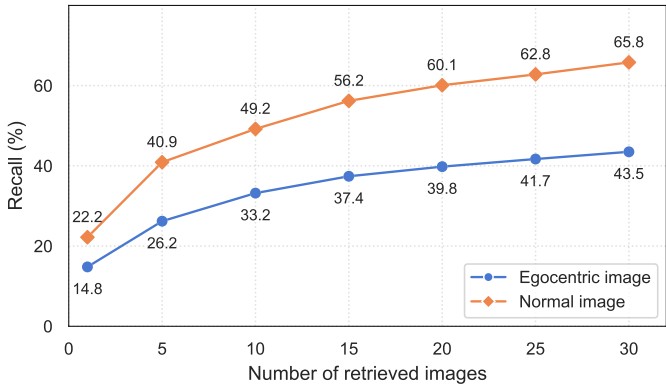

Figure 7: Image search recall by image type using original image as query images.

### A.3.3 WEB SEARCH API CREATION

We created the webpage repository in three steps.

1. We created the ground truth search query and some negative search queries. We constructed the ground truth query by asking the annotators to write down a standalone query for answering the given question regarding the image, without the need to use the image. For example, "who wrote the Martian" for "who wrote this book". To simulate the noise, we also created negative search queries in two ways: 1) a query including the ground truth entity only, 2) queries including similar entities to the ground truth entity. For example, we use Toyota Rav4 if the question is about Volkswagon Tiguan.

2. We next used each query to search and store up to 20 urls via the Brave search API Software. We collected up to 50 distinct webpages for each question by combining the search results returned by all the queries and retained the top 50.

3. We then pooled the webpages for all questions together and constructed a mock web search API using ChromaDB.

### A.3.4 UNIFIED SEARCH API

We built a light-weight Python package, providing a unified API to support image and web search. The API can take image URL, PIL image and text query and decide which index (image or web search index) to query depending on the input data type. Below are examples of how to use the search API.

```
# image search example
results = search_pipeline(image, k = 1)
for result in results:
    print(result)
    print('\n')
# output
{'index': 17030, 'score': 0.906402587890625, 'url': 'https://upload.
    wikimedia.org/wikipedia/commons/3/34/
    The_Beekman_tower_from_the_East_River_%286215420371%29.jpg', '
    entities': [{'entity_name': '8 Spruce Street', 'entity_attributes':
    {'name': '8 Spruce Street
(New York by Gehry)', 'image': '8
    Spruce Street (01030p).jpg', 'image_size': '200px', 'address': '8
    Spruce Street
[[Manhattan]], New York, U.S. 10038', '
    mapframe_wikidata': 'yes', 'coordinates': '{{coord|40|42|39|N
    |74|00|20|W|region:US-NY_type:landmark|display|=|inline,title}}', '
    status': 'Complete', 'start_date': '2006', 'completion_date': '2010',
     'opening': 'February 2011', 'building_type': '[[Mixed-use
    development|Mixed-use]]', 'architectural_style': '[[Deconstructivism
    ]]', 'roof': '{{convert|870|ft|m|0|abbr|=|on}}', 'top_floor': '{{
    convert|827|ft|abbr|=|on}}', 'floor_count': '76', 'floor_area': '{{
    convert|1000000|sqft|m2|abbr|=|on}}', 'architect': '[[Frank Gehry]]',
     'structural_engineer': '[[WSP Group|WSP Cantor Seinuk]]', '
    main_contractor': 'Kreisler Borg Florman', 'developer': '[[Forest
    City Ratner]]', 'engineer': '[[Jaros, Baum & Bolles]] (MEP)', 'owner
    ': '8 Spruce (NY) Owner LLC', 'management': 'Beam Living', 'website':
     '{{URL|https://live8spruce.com/}}'}}]}

# web search example (results partially shown)
query='What to know about Andrew Cuomo?'
results = search_pipeline(query, k=1)
for result in results:
    print(result)
    print('\n')
# output
{'index': 'https://en.wikipedia.org/wiki/Mario_Cuomo_chunk_2', 'score':
    0.5727531909942627, 'page_name': 'Mario Cuomo - Wikipedia', '
    page_snippet': 'He vigorously attacked Ronald Reagan's ...
    brought him to national attention, most memorably saying: "There
     is despair, Mr. President, in the faces that you don't see, in
    the places that you don't visit, in your shining city." He
    was immediately considered one of the frontrunners for the Democratic
     ...He vigorously attacked Ronald Reagan\'s ... brought ......', '
    page_url': 'https://en.wikipedia.org/wiki/Mario_Cuomo'}
```

### A.4 AUTO-EVALUATION WITH LLM-AS-A-JUDGE

We used GPT-4o to evaluate whether the responses are correct, hallucinated or missing, given the queries and ground truth answers. To ensure that the auto-evaluator produces accurate and reliable judgments, we implemented multiple layers of quality control for both the ground-truth answers and the validation procedure. As shown in Tab. 7, the judge achieves 99% average accuracy and 91% average F1 score across the three categories.

We detail below the steps taken to ensure the accuracy of LLM-based evaluation.

Table 7: Performance of the GPT-4o based auto-evaluation judge.

|  | Accuracy | Precision | Recall | F1 score |
|---|---|---|---|---|
| **Accurate** | 98.8 | 98.1 | 90.8 | 94.3 |
| **Hallucinated** | 98.7 | 68.1 | 91.7 | 78.2 |
| **Missing** | 100.0 | 100.0 | 100.0 | 100.0 |
| **Average** | 99.1 | 88.7 | 94.2 | 90.8 |

**High-quality ground-truth answers.** To reduce evaluation ambiguity, we curated and validated high-quality ground-truth answers so that auto-evaluation is reduced to a semantic-matching task. Our process included three quality-control stages:

1. **Qualification and annotation.** All annotators received training and were required to pass a qualification test for the grading guidelines.

2. **Initial data creation.** During QA construction, one annotator verified the factual correctness of each answer and made correction if needed.

3. **Review and expert audit.** A second annotator conducted a full independent review, followed by a sampled audit by an expert annotator. Approximately 7% of answers were corrected during this stage, resulting in a final ground-truth correctness rate of ∼95%.

**Validation of the LLM-judge.** To ensure the reliability of the human labels used for validating the LLM-judge, we performed the following steps:

1. **Diverse data collection.** We randomly sampled responses from both baseline MM-RAG systems and public challenge submissions across all tasks (12k / 7k / 5k samples for Tasks 1–3), minimizing sampling bias.

2. **Qualification and annotation.** Annotators passed two qualification stages, one on task objectives and one on the grading guideline, and evaluated responses for factual correctness, relevance, and contextual reasoning following strict guidelines. In detail, qualified annotators were given an image, query, ground truth, conversation history if applicable and a model response. The task was to evaluate the model response quality against the ground truth, focusing strictly on accuracy, relevance and contextual understanding. The guideline enforces a two-step process: first, decide if the job needs to be rejected if it contains privacy risks, safety violations, unanswerable due to visual quality or non-exhaustive queries where multiple valid answers exist. Secondly, valid responses were then graded on a 1-3 scale: 1 (Major Hallucination/Irrelevant), 2 (Acceptable with Minor Non-harmful Hallucination), and 3 (Accurate & Complete). Note that human grading is not needed for "missing" cases.

3. **Expert audit.** In-house linguists audited a random sample of the annotations, achieving 92% inter-annotation agreement and confirming 88% / 91% quality for single-turn and multi-turn response evaluation.

**Evaluation prompt.** We carefully crafted in-context learning examples and tuned the prompt used for the evaluation. We include ∼20 in-context examples to improve the judge performance but only show one example here due to space constraints.

```
Auto-evaluation prompt

You will be given a question, a ground truth answer, and a model
    prediction. Your task is to judge if the prediction is correct or
    not based on the ground truth answer.

## Instructions
Read the question, ground truth answer, and model prediction carefully.
    Follow the step by step guideline below to make a judgment.
1. If the prediction indicates uncertainty or refusal to answer, output
    "Result: WRONG".
```

```
2. If the prediction exactly matches the ground truth, output "Result:
   CORRECT".
3. If the ground truth is a number
    3.1 If the prediction gives a number that almost exactly matches
    the ground truth, output "Result: CORRECT".
    3.2 If the prediction gives a number that is not the same as the
    ground truth, output "Result: WRONG".
4. If the prediction is self-contradictory, output "Result: WRONG".
5. If the prediction is not answering the question, output "Result:
   WRONG".
6. If ground truth contains a set of objects,
    6.1 if the prediction contains exactly same objects as the ground
    truth, output "Result: CORRECT".
    6.2 if the prediction contains different objects from the ground
    truth, output "Result: WRONG".
    6.3 if the prediction is almost same as the ground truth, use your
    best judgement to give output.
7. If the prediction is grounded by the ground truth, output "Result:
   CORRECT".
8. If the prediction is unrelated or contradictory to the ground truth,
    output "Result: WRONG".
## Additional Guidelines
- Take it as granted that the ground truth is always correct.
- If the prediction gives extra information that is not in the ground
    truth, it is still correct as long as it is grounded by the ground
    truth.
- Be careful about numbers. 1 mile is about 1.60934 km. 1 foot is about
    0.3048 m. 1 inch is about 2.54 cm. 1 yard is about 0.9144 m. 1
    pound is about 0.453592 kg. 1 gallon is about 3.78541 liters. 1
    ounce is about 28.3495 grams.

## Output Format
Your judgment should first provide a VERY-SHORT explanation on your
    rationale. When relevant, you need to include the guidelines above
    to explain your judgment. Finally, your judgment should clearly
    state "Result: CORRECT" or "Result: WRONG".

Below are some examples:
EXAMPLES START
Question: who will win the game?
Ground Truth: Lakers is favored to win the game.
Prediction: Sorry, it is hard to predict the outcome of the game.
Explanation: The prediction indicates it is not sure about the answer.
    So the prediction is incorrect according to the guideline 1.
Result: WRONG
. . .
EXAMPLES END
```

To encourage concise answers, we truncated the answers to 75 tokens.

## A.5 EVALUATING STRAIGHTFORWARD MM-RAG SOLUTIONS

### A.5.1 IMPLEMENTATION DETAILS

**Single-source augmentation (Task 1).** We leverage an image search API to retrieve visually similar images and their associated metadata, providing external knowledge to support entity recognition and answer generation, particularly for torso-to-tail entities that MM-LLMs often hallucinate. In our baseline implementation, we use the full input image to query the image search API and retrieve the top 30 images and metadata. A similarity threshold of 0.75 is applied to filter noisy results. The retrieved entity names are retained, and the associated metadata is truncated to 2k tokens. The resulting image search content is appended to the model prompt for response generation.

**Multi-source augmentation (Task 2 and 3).** We incorporate web content in addition to image search, implementing a multi-stage pipeline that integrates image search, query rewriting, web search, and answer generation: Specifically:

1. We first retrieve the top 30 entities and associated metadata (as in the single-source setting).

2. Given the input image, query, and retrieved content, we use Llama-3.2-11B-Vision-Instruct to rewrite the original query into a fully specified textual query independent of the image.

3. The rewritten query is used to query the web search API, returning up to 50 relevant webpages.

4. We construct the final prompt by concatenating the image search content (up to 2k tokens), web snippets (up to 8k tokens), and the query.

5. For multi-turn QA, we additionally include the full conversation history, with previous responses generated by the agent under evaluation.

### A.5.2 PROMPT CONSTRUCTION

We show our prompts used to produce the evaluation results in Tab. 4, Tab. 8 and Tab. 9. All the models or APIs are evaluated under the same prompt construction format.

---

**Prompt template for MM-LLM only solution, single-turn QA**

```
<|system|>
You are a helpful assistant that truthfully answers the user question
    given an image.
Please follow these guidelines when formulating your response:
1. Your response must be grounded in the image and based on factual
    information.
2. Keep your response concise and to the point. Strive to answer in one
    sentence.
3. If you are uncertain or don't know the answer, respond with "I don't
    know".
<|user|>
Image: <image>
Query: <query>
```

---

**Prompt template for MM-LLM only solution, multi-turn QA**

```
<|system|>
You are a helpful assistant that truthfully answers the user question
    given an image. You are able to engage in multi-turn conversations.
     Answer the new question based on the given image and information
    extracted from the previous conversations.
Please follow these guidelines when formulating your response:
1. Your response must be grounded in the image and based on factual
    information.
2. Build upon previous conversations when responding.
3. Keep your response concise and to the point. Strive to answer in one
    sentence.
4. If you are uncertain or don't know the answer, respond with "I don't
    know".
<|user|>
Image: <image>
Conversation history: <conversation_history>
Query: <query>
```

**Prompt template for single-turn augmentation**

```
<|system|>
You are a helpful assistant that truthfully answers the user question
    given an image.
Please follow these guidelines when formulating your response:
1. Your response must be grounded in the image and based on factual
    information.
2. Keep your response concise and to the point. Strive to answer in one
    sentence.
3. If you are uncertain or don't know the answer, respond with "I don't
    know".
<|user|>
Image: <image>
Image search context:
Here is a list of entity names retrieved based on visual similarity to
    the provided image:
<list_of_entity_names>
Here are some additional attributes for some of the entities. Only
    incorporate this information into your answer ONLY IF you are
    confident the referenced entity is in the provided image.
Entity : <entity_name>
Entity attributes: <meta_data>
. . .
Entity <n>: <entity_name>
Entity attributes: <meta_data>
Incorporate these image entity information into your answer ONLY IF you
    are confident they refer to the exact same entity. Disregard them
    otherwise.
Query: <query>
```

**Prompt template for multi-source augmentation**

```
<|system|>
You are a helpful assistant that truthfully answers the user question
    given an image.
Please follow these guidelines when formulating your response:
1. Your response must be grounded in the image and based on factual
    information.
2. Keep your response concise and to the point. Strive to answer in one
    sentence.
3. If you are uncertain or don't know the answer, respond with "I don't
    know".
<|user|>
Image: <image>
Image search context:
Here is a list of entity names retrieved based on visual similarity to
    the provided image:
<list_of_entity_names>
Here are some additional attributes for some of the entities. Only
    incorporate this information into your answer ONLY IF you are
    confident the referenced entity is in the provided image.
Entity : <entity_name>
Entity attributes: <meta_data>
. . .
Entity <n>: <entity_name>
Entity attributes: <meta_data>
Incorporate these image entity information into your answer ONLY IF you
    are confident they refer to the exact same entity. Disregard them
    otherwise.
```

```
You are given snippets from web page search results based on this
    question: "{query}". These page snippets may or may not contain
    relevant or truthful information about the question. Incorporate
    these information into your answer ONLY IF you are confident they
    address the question. Disregard them otherwise.
Web search context:
<DOC>
Webpage ID: 
Title: <page_name>
Web content snippet: <snippet>
</DOC>
Incorporate these web search results into your answer ONLY IF you are
    confident they contain relevant or truthful information about the
    question. Disregard them otherwise.
Query: <query>
```

## A.6 PERFORMANCE OF STRAIGHTFORWARD SOLUTIONS

In addition to the models presented in Section 5.2.1, we evaluated our straightforward MM-RAG solutions on CRAG-MM with the following MM-LLMs for a more comprehensive performance comparison: `Llama-4-Maverick` (Meta AI, 2025), `Pixtral-12B-2049` (Agrawal et al., 2024), `Qwen-2.5-VL-72B` (Bai et al., 2025), `InternVL3.5-38B` (Wang et al., 2025) and `Claude 3.7 Sonnet` (Anthropic, 2025). Tab. 8 and Tab. 9 present the evaluation results for single-turn and multi-turn QAs respectively.

Table 8: Performance of straightforward MM-RAG solutions on CRAG-MM single-turn QA.

| Single-turn | Model | Acc. | Miss. | Hallu. | Truth. |
|---|---|---|---|---|---|
| **MM-LLM** | Llama 3.2 11B | 24.4 | 34.4 | 41.3 | -16.9 |
| | Llama 3.2 90B | 28.2 | 39.1 | 32.8 | -4.6 |
| | Llama 4 Maverick (FP8) | 33.3 | *11.0* | 55.7 | -22.4 |
| | Pixtral 12B 2409 | 21.9 | 36.1 | 42.1 | -20.1 |
| | Qwen2.5 VL 72B | 26.0 | 45.9 | 28.1 | -2.1 |
| | InternVL3.5 38B | 18.0 | 54.7 | 26.6 | -7.9 |
| | Claude 3.7 Sonnet | 30.9 | 28.9 | 40.2 | -9.3 |
| | Gemini 2.5 Flash | 36.6 | 38.0 | 25.4 | 11.2 |
| | GPT-5 Mini | *37.4* | 43.7 | *19.0* | *18.4* |
| **Task 1** | Llama 3.2 11B | 19.0 | 52.4 | 28.7 | -9.7 |
| | Llama 3.2 90B | 13.5 | 71.0 | 15.6 | -2.1 |
| | Llama 4 Maverick (FP8) | 23.5 | 52.5 | 24.0 | -0.5 |
| | Pixtral 12B 2409 | 23.1 | 35.2 | 41.6 | -18.5 |
| | Qwen2.5 VL 72B | 20.3 | 63.1 | 16.6 | 3.7 |
| | InternVL3.5 38B | 13.4 | 74.2 | **12.4** | 1.0 |
| | Claude 3.7 Sonnet | 24.5 | *21.5* | 54.0 | -29.5 |
| | Gemini 2.5 Flash | 37.9 | 36.8 | 25.3 | 12.6 |
| | GPT-5 Mini | *39.3* | 43.9 | 16.8 | *22.5* |
| **Task 2** | Llama 3.2 11B | 35.3 | 20.8 | 43.9 | -8.6 |
| | Llama 3.2 90B | 30.1 | 50.1 | 19.8 | 10.3 |
| | Llama 4 Maverick (FP8) | 44.1 | 15.0 | 40.9 | 3.2 |
| | Pixtral 12B 2409 | 42.1 | 4.9 | 53.1 | -11.0 |
| | Qwen2.5 VL 72B | 31.8 | 44.7 | 23.6 | 8.2 |
| | InternVL3.5 38B | 37.9 | 34.3 | 27.8 | 10.1 |
| | Claude 3.7 Sonnet | 30.7 | **0.9** | 68.4 | -37.7 |
| | Gemini 2.5 Flash | **49.9** | 22.6 | 27.5 | 22.4 |
| | GPT-5 Mini | 48.7 | 34.1 | *17.2* | **31.5** |

Table 9: Performance of straightforward MM-RAG solutions on CRAG-MM multi-turn QA.

| Multi-turn | Model | Acc. | Miss. | Hallu. | Truth. | Early Stop. |
|---|---|---|---|---|---|---|
| **MM-LLM** | Llama 3.2 11B | 36.5 | 19.5 | 44.0 | 1.6 | 74.9 |
| | Llama 3.2 90B | 42.2 | 25.0 | 32.8 | 12.7 | 64.7 |
| | Llama 4 Maverick (FP8) | 43.4 | 16.0 | 40.6 | 9.8 | 65.7 |
| | Pixtral 12B 2409 | 27.9 | 44.2 | 27.9 | 2.5 | 82.3 |
| | Qwen2.5 VL 72B | 40.1 | 25.6 | 34.3 | 12.1 | 71.3 |
| | InternVL3.5 38B | 37.6 | *9.4* | 53.0 | -1.1 | 73.0 |
| | Claude 3.7 Sonnet | 38.2 | 30.2 | 31.6 | 10.9 | 74.6 |
| | Gemini 2.5 Flash | 29.2 | 57.5 | **13.4** | 16.5 | 88.1 |
| | GPT-5 Mini | *48.9* | 34.0 | 17.1 | *30.4* | *60.8* |
| **Task 3** | Llama 3.2 11B | 46.6 | 11.0 | 42.4 | 8.8 | 63.7 |
| | Llama 3.2 90B | 37.1 | 46.1 | 16.9 | 18.9 | 81.7 |
| | Llama 4 Maverick (FP8) | 53.9 | 9.5 | 36.6 | 19.7 | 55.5 |
| | Pixtral 12B 2409 | 48.7 | 6.5 | 44.7 | 9.8 | 62.3 |
| | Qwen2.5 VL 72B | 42.0 | 30.2 | 27.9 | 17.0 | 71.0 |
| | InternVL3.5 38B | 50.3 | 16.9 | 32.8 | 20.5 | 58.0 |
| | Claude 3.7 Sonnet | 55.4 | **5.0** | 39.6 | 19.6 | 49.7 |
| | Gemini 2.5 Flash | 54.4 | 24.2 | 21.4 | 31.4 | 55.8 |
| | GPT-5 Mini | **61.0** | 22.5 | *16.5* | **42.5** | **43.5** |

### A.6.1 PERFORMANCE OF STRAIGHTFORWARD SOLUTIONS ON CRAG-MM SLICES

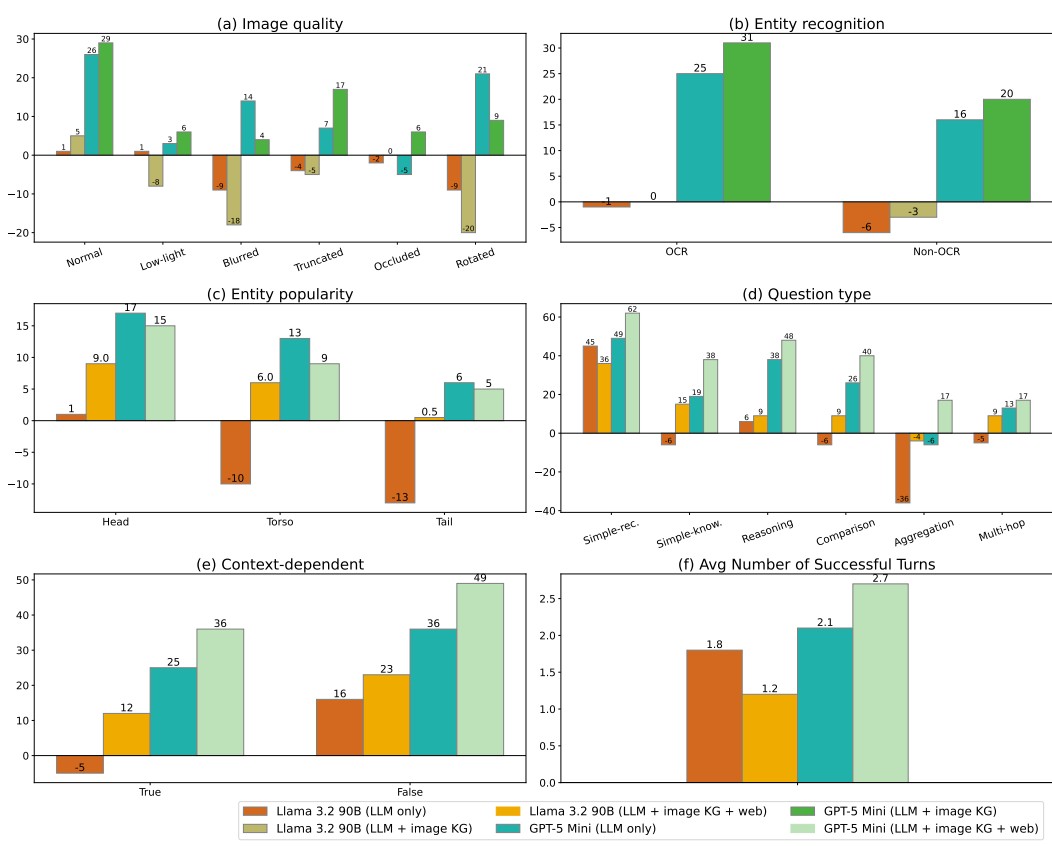

Figure 8: Performance of straightforward MM-RAG solutions across different slices of CRAG-MM. Figure (a)–(e) show truthfulness in percentage; figure (f) shows the average number of successful turns in multi-turn QA.

Fig.8 shows the performance of the MM-LLM-only and straightforward solutions on different slices of CRAG-MM, showing that CRAG-MM reveals interesting insights and allows large room for improvement for developing MM-RAG systems.

First, understanding the entity and context in the image is important and challenging for wearable QA, particularly when there is quality issue in the image (Fig.8a) or when text is not present - the model needs to recognize the entity purely from the visual information (Fig.8b). Adding image search in a crude manner shows mixed effect on different low quality categories (-11% - +10%), showing the need for proper pre-processing or smarter image search.

Second, properly leveraging the retrieved information is non-trivial, particularly when the query involves entities that are less popular (Fig.8c), requires external knowledge, or demands synthesizing multiple pieces of information to produce an answer (Fig.8d).

Third, results on multi-turn conversation reveals a large gap in conducting smooth conversation - more than 44% conversations were early stopped (has two consecutive failures, Tab.9); the best truthfulness score is only 36% for questions that require conversation history to answer (Fig.8e); on average the best system only has 2.7 successful turns among the 4.9 total for each multi-turn conversation (Fig.8f).

### A.6.2 PERFORMANCE OF STRAIGHTFORWARD SOLUTIONS ON QUERIES WITH IMAGE SEARCH COVERAGE

Table 10: Performance of straightforward MM-RAG solutions on a subset of 65% single-turn QAs in CRAG-MM, where queries that failed to retrieve the correct facilitating information were removed. Even when the answer exists in the retrieval results, the truthfulness is up to only 39.3%.

| Single-turn | Model | Acc. | Miss. | Hallu. | Truth. |
|---|---|---|---|---|---|
| **MM-LLM** | Llama 3.2 11B | 26.8 | 31.1 | 42.1 | -15.3 |
| | Llama 3.2 90B | 30.3 | 37.4 | 32.3 | -2.1 |
| | Llama 4 Maverick (FP8) | 36.3 | *11.5* | 52.2 | -16.0 |
| | Pixtral 12B 2409 | 23.9 | 33.3 | 42.7 | -18.8 |
| | Qwen2.5 VL 72B | 29.9 | 41.1 | 28.7 | 1.3 |
| | InternVL3.5 38B | 21.2 | 52.4 | 26.4 | -5.1 |
| | Claude 3.7 Sonnet | 32.7 | 27.6 | 39.7 | -7.0 |
| | Gemini 2.5 Flash | 39.3 | 36.5 | 24.2 | 15.0 |
| | GPT-5 Mini | *39.7* | 42.5 | *17.9* | *21.8* |
| **Task 1** | Llama 3.2 11B | 23.4 | 47.2 | 29.5 | -6.1 |
| | Llama 3.2 90B | 16.8 | 67.1 | 16.1 | 0.7 |
| | Llama 4 Maverick (FP8) | 29.0 | 44.8 | 26.2 | 2.8 |
| | Pixtral 12B 2409 | 27.0 | 30.6 | 42.4 | -15.4 |
| | Qwen2.5 VL 72B | 24.7 | 55.7 | 19.6 | 5.1 |
| | InternVL3.5 38B | 17.5 | 68.2 | **14.3** | 3.2 |
| | Claude 3.7 Sonnet | 29.0 | *19.9* | 51.1 | -22.1 |
| | Gemini 2.5 Flash | 42.2 | 34.6 | 23.2 | 19.0 |
| | GPT-5 Mini | *43.5* | 40.4 | 16.0 | *27.5* |
| **Task 2** | Llama 3.2 11B | 39.5 | 19.4 | 41.2 | -1.7 |
| | Llama 3.2 90B | 34.3 | 46.0 | 19.7 | 14.5 |
| | Llama 4 Maverick (FP8) | 49.8 | 13.0 | 37.2 | 12.6 |
| | Pixtral 12B 2409 | 46.8 | 5.1 | 48.1 | -1.3 |
| | Qwen2.5 VL 72B | 36.5 | 39.1 | 24.4 | 12.1 |
| | InternVL3.5 38B | 42.8 | 31.1 | 26.1 | 16.7 |
| | Claude 3.7 Sonnet | 34.8 | **0.7** | 64.5 | -29.6 |
| | Gemini 2.5 Flash | **55.5** | 20.1 | 24.4 | 31.0 |
| | GPT-5 Mini | 55.1 | 29.1 | *15.8* | **39.3** |

Table 11: Performance of straightforward MM-RAG solutions on a subset of 85% multi-turn sessions in CRAG-MM, where sessions that failed to retrieve the correct facilitating information were removed. Even when the answer exists in the retrieval results, the truthfulness is up to only 42.4%.

| Multi-turn | Model | Acc. | Miss. | Hallu. | Truth. | Early Stop. |
|---|---|---|---|---|---|---|
| **MM-LLM** | Llama 3.2 11B | 36.5 | 18.9 | 44.2 | 2.0 | 73.9 |
| | Llama 3.2 90B | 42.4 | 24.8 | 32.8 | 12.9 | 64.2 |
| | Llama 4 Maverick (FP8) | 42.9 | 17.8 | 39.3 | 9.7 | 67.1 |
| | Pixtral 12B 2409 | 27.6 | 45.1 | 27.3 | 2.9 | 82.6 |
| | Qwen2.5 VL 72B | 39.5 | 26.6 | 33.9 | 12.6 | 71.3 |
| | InternVL3.5 38B | 37.5 | *9.7* | 52.8 | -1.5 | 73.7 |
| | Claude 3.7 Sonnet | 37.3 | 32.4 | 30.3 | 11.0 | 75.4 |
| | Gemini 2.5 Flash | 28.1 | 59.0 | **12.9** | 16.2 | 87.9 |
| | GPT-5 Mini | *47.4* | 36.3 | 16.3 | *29.8* | *62.8* |
| **Task 3** | Llama 3.2 11B | 48.0 | 10.9 | 41.0 | 11.0 | 61.6 |
| | Llama 3.2 90B | 38.8 | 44.1 | 17.0 | 20.4 | 71.7 |
| | Llama 4 Maverick (FP8) | 55.0 | 9.8 | 35.3 | 21.9 | 52.9 |
| | Pixtral 12B 2409 | 49.8 | 7.1 | 43.1 | 11.4 | 59.4 |
| | Qwen2.5 VL 72B | 42.6 | 30.1 | 27.3 | 17.8 | 70.3 |
| | InternVL3.5 38B | 51.8 | 17.1 | 31.1 | 22.3 | 56.4 |
| | Claude 3.7 Sonnet | 56.6 | **5.2** | 38.2 | 21.0 | 48.7 |
| | Gemini 2.5 Flash | 55.4 | 24.1 | 20.6 | 32.8 | 54.5 |
| | GPT-5 Mini | **60.9** | 22.9 | *16.2* | **42.4** | **44.2** |

## A.7 EVALUATING INDUSTRY SOTA SOLUTIONS

### A.7.1 EXPERIMENTS

We tested the SOTA MM-RAG systems by sending each question in the public test set as input and collecting the responses for evaluation. Since these SOTA systems all have embedded retrieval systems, the provided retrieval results in CRAG-MM were **not** used.

We evaluated three systems: GPT-5-2025-08-07, Gemini-2.5-Pro and Claude-Sonnet-4-20250514, and assessed via the API platforms provided by OpenAI, Google AI and Anthropic, respectively. Tool usage was explicitly enabled to ensure search functionality was available for our end-to-end assessment of MM-RAG systems. Each system was called during the following dates in Pacific Time: 09/01/2025~09/09/2025 (GPT-5), 09/01/2025~09/10/2025 (Gemini), and 09/02/2025~09/10/2025 (Claude Sonnet).

Below is our parameter setting for tool usage specifically.

```
gpt_tool_calling = [
    {
        "type": "web_search_preview"
    }
]
gemini_generate_content_config = types.GenerateContentConfig(
    thinking_config=types.ThinkingConfig(
        thinking_budget=-1,
    ),
    tools= [
        types.Tool(googleSearch=types.GoogleSearch()),
    ],
)
claude_tool_calling = [
        {
            "name": "web_search",
            "type": "web_search_20250305"
        }
]
```

We show the prompts used to produce the SOTA MM-RAG evaluation results in Tab.5 and Fig.4 as follows. The prompts used in the evaluation are shown in Table 1.

---

**Prompt template for single-turn QA**

```
You are a helpful assistant that truthfully answers user questions
    about the provided image.
Keep your response concise and to the point. Strive to answer in one
    sentence.
If you are unsure or don't know the answer, respond with "I don't know
    ".
Search the internet and provide a short answer to the question: <query>
Image: <image>
```

---

**Prompt template for multi-turn QA**

```
Conversation history: <conversation_history>
You are a helpful assistant that truthfully answers user questions
    about the provided image.
Keep your response concise and to the point. Strive to answer in one
    sentence.
If you are unsure or don't know the answer, respond with "I don't know
    ".
Search the internet and provide a short answer to the question: <query>
Image: <image>
```

