# OpenReview forum: "CRAG-MM: Multi-modal Multi-turn Comprehensive RAG Benchmark"
_ICLR.cc/2026/Conference — Submitted to ICLR 2026_

### Official Review · Reviewer_qUKK · 2025-10-26

**Soundness:** 2
**Presentation:** 2
**Contribution:** 2
**Rating:** 4
**Confidence:** 3

**Summary:**

The paper introduces CRAG-MM, a new benchmark for evaluating Multi-Modal Retrieval-Augmented Generation (MM-RAG) systems. The authors motivate this work by highlighting a critical gap in modern wearable AI devices which need to answer factual, multi-turn questions about a user's visual surroundings. This benchmark fill this gap by building with egocentric images. The paper proposes three evaluation tasks of increasing complexity: single-source augmentation (retrieval from an image-based Knowledge Graph), Multi-source augmentation (retrieval from both image-KG and a webpage corpus, and Multi-turn QA (using retrieval and conversation history). Benchmarking experiments show that even state-of-the-art industry RAG solutions perform poorly, achieving only 32% (single-turn) and 45% (multi-turn) truthfulness.

**Strengths:**

1. The paper identifies a clear and important gap in existing research.
2. This paper provide a novel new benchmark.

**Weaknesses:**

1. Although a novel benchmark, this paper does not offer significant analysis or interesting insights from its benchmark.
2. The paper lacks of a methodology of how to improve the model's capabilities.
3. Is Truthfulness = Accuracy - Hallucination? It's not consistent in the paper.

**Questions:**

n/a

---

> ### Author Response · Authors · 2025-11-22
> **CRAG-MM Reveals Novel and Practically Important Findings**
>
> Re: "Although a novel benchmark, this paper does not offer significant analysis or interesting insights from its benchmark."
>
> We believe CRAG-MM reveals several **previously unobserved and practically important findings** about MM-RAG systems in real-world, egocentric settings:
>
> 1. **Commercial MM-RAG systems do not outperform straightforward baselines.** A key insight from our analysis is that widely deployed commercial MM-RAG systems perform on par with a simple straightforward RAG pipeline. Truthfulness remains low across all systems (e.g., ~32% vs. 32% on single-turn and 43% vs. 45% on multi-turn), despite the commercial systems’ stronger proprietary retrievers and larger corpora. This highlights a **surprising robustness gap** between performance on clean VQA benchmarks and real-world wearable/egocentric scenarios, where blur, occlusion, low light, and long-tail entities are common. To our knowledge, no prior benchmark has surfaced these findings.
>
> 2. **Retrieval alone does not resolve hallucination, even for strong VLMs.**
> Our controlled comparisons show that a strong MM-LLM (GPT-5-mini) achieves only 37% accuracy without retrieval. While adding RAG improves accuracy, hallucination remains substantial for both straightforward solutions and commercial systems (e.g., 31% hallucination for GPT-5 with retrieval). This demonstrates that **hallucination persists even with high-capacity models and high-quality web search**, indicating that retrieval is not a silver bullet for multimodal grounding in challenging, real-world visual contexts.
>
> 3. **Careful optimizations for retrieval pipeline quality and generation model fine-tuning can significantly reduce hallucinations, whereas accuracy improvement call for further retrieval improvement.** As shown in Table 5, winning team solutions reduced hallucination rate by ~35%, whereas maintained similar accuracy.
>
> 4. Despite the significant improvement of long-context generation, **RAG quality still reduces significantly from single-turn to multi-turn conversations**. Figure 4 shows that truthfulness for context-dependent turns is +10% higher than that for context-independent turns even from SOTA industry solutions, and Figure 8 illustrates the same finding on straightforward solutions.

---

> ### Author Response · Authors · 2025-11-23
> **Clarifying Evaluation Metrics**
>
> Re: "Is Truthfulness = Accuracy - Hallucination? It's not consistent in the paper."
>
> We would like to clarify that we use a consistent definition across the paper, and we will revise the text to avoid ambiguity.
>
> For **single-turn QA**, Truthfulness = Accuracy − Hallucination.
>
> For **multi-turn conversations**, each turn is scored using the same definition above. The only difference is **early stopping**: if a model produces two consecutive hallucinated or missing answers, we terminate the conversation and label all remaining turns as **missing**. This mimics realistic user behavior—users typically stop interacting with a system after repeated failures. The **conversation-level Truthfulness** is then computed as the average Truthfulness across all the turns.

---

> ### Author Response · Authors · 2025-11-24
> **Clarifying Scope: Benchmarking as a Foundation for Methodological Advancement**
>
> Re: "The paper lacks of a methodology of how to improve the model's capabilities."
>
> We thank the reviewer for this comment. We respectfully clarify that the primary contribution of this work is to **bridge a significant gap** in the community by introducing the first benchmark (to our best knowledge) specifically designed to evaluate MM-RAG systems for Wearable scenarios. In the current landscape, the lack of a specialized benchmark has been a major bottleneck for methodological progress.
>
> While proposing a new methodology is not the focus of this paper, we believe CRAG-MM provides the foundation for developing and comparing such methods. Specifically, it drives improvements through:
>
> - **Defining the problem space:** The benchmark captures unique challenges,  such as long-tail entity recognition, noisy egocentric imagery, multi-step search, and hallucination under uncertainty, that existing RAG benchmarks do not represent.
> - **Incentivizing reliable solutions:** We adopt the evaluation metric that assigns a penalty to hallucinations while treating abstention neutrally. This aligns with recent theoretical findings in [1] that commonly adopted evaluation with binary grading inadvertently lead to hallucinate by rewarding guessing over uncertainty. By explicitly penalizing incorrect answers more than missing ones, CRAG-MM can incentivize the development of RAG systems that are trustworthy and honest about their own limitations.
> - **Diagnosing failure modes:** Our analysis reveals that current SOTA MM-RAG systems struggle with long-tail entities and quries that require multi-turn context. Exposing these specific performance gaps is a necessary step for targeted model capability improvement.
> - **Enabling rapid iteration:** As demonstrated by our public challenge, CRAG-MM provides the standardized framework essential for method development. The competition has already yielded promising approaches, including improved abstention strategies and robust entity recognition for low-quality egocentric images, demonstrating the benchmark’s utility in catalyzing new research.
>
> [1] https://arxiv.org/abs/2509.04664

---

### Official Review · Reviewer_kBjy · 2025-10-29

**Soundness:** 3
**Presentation:** 3
**Contribution:** 4
**Rating:** 6
**Confidence:** 2

**Summary:**

This paper introduces CRAG-MM, a Comprehensive RAG benchmark for Multi-modal Multi-turn conversations.
The dataset contains a diverse set of 6.5K (image, question, answer) triplets and 2K multi-turn visual conversations across 13 domains, including 6.2 K egocentric images from wearable device smart glasses.
The benchmark emphasizes real-world challenges like low-light, blur, truncation, occlusion, and rotation, reflecting the conditions faced in wearable AI applications. It supports three tasks (single-source, multi-source, and multi-turn RAG). Experiments show that even advanced systems (e.g., GPT-5) achieve only ~63 % (single-turn) and ~70 % (multi-turn) truthfulness, indicating room for progress. A public leaderboard has drawn ~1 K participants, showing early community engagement and progress on this benchmark.

**Strengths:**

1) Rich conversational coverage: Includes 2 K multi-turn conversations, ∼38 % of which involve domain shifts, realistically simulating natural topic drift.

2) Real-world visual realism: Contains 7.9 K images, with 79 % egocentric, capturing wearable AI’s inherent visual challenges (wide-angle, occlusion, low light).

3) Comprehensive evaluation: GPT-5 achieves 63 % (single-turn) and 70 % (multi-turn) accuracy, revealing a measurable gap and potential for improvement on MM-RAG.

4) Community impact: The dataset powered a leaderboard competition that attracted ~1 K participants.

5) Human involvement: Data were created or revised by human annotators to ensure quality and realistic question–answer alignment.

6) Extensive analysis: The paper includes exploratory data analysis, distributions by image quality, domain, and question type, supporting interpretability and dataset transparency.

**Weaknesses:**

Ethical and safety concerns: The dataset involves vendors wearing smart glasses in daily contexts. While this enables realism, it raises potential privacy and identifiability risks for bystanders. Clearer documentation of anonymization and consent protocols is needed.

**Questions:**

Please clarify the licensing and privacy protections for public images and web pages, as well as annotator instructions regarding bystanders.
Were faces or license plates blurred or removed before release?
How are data from wearable captures handled to ensure participant safety and privacy compliance?

**Details Of Ethics Concerns:**

The dataset involves vendors wearing smart glasses in daily contexts. While this enables realism, it raises potential privacy and identifiability risks for bystanders.

---

> ### Author Response · Authors · 2025-11-22
> **Ethical Compliance: Privacy Protection, Consent Protocols, and Data Licensing**
>
> We thank the reviewer for raising this important point. We took multiple steps to ensure that privacy and consent requirements were strictly followed.
>
> - All images were collected by vendors who signed an agreement. The agreement explicitly states “You will only use the Products within your home and approved outdoor locations and will not allow any other person to be recorded by the Products unless such person is a household member who is at least 18 years of age and has signed a separate household member release (“Household Member Release”).” We also explicitly instructed the vendors **not to capture people or privacy-sensitive information** (e.g., IDs, phone numbers, emails, etc).
>
> - In addition, we followed a rigorous and extensive process to obtain the privacy approval for data release. As part of this process, all identifiable faces were blurred to ensure privacy.
>
> - Finally, we have carefully reviewed each image and removed the ones with any sensitive information or safety violation. This includes privacy-sensitive contents or topics related to child abuse, dangerous organizations, hate speech, mental health, and politics.
>
> **Licensing for public images and web pages.**
>
> We applied rigorous criteria to ensure license and privacy requirements are met. We only include public images with license that permits research use (e.g. CC-BY-4.0). Also, the released data only contain the url not the public images or web page contents.
>
> We will clarify this in the revised manuscript.

---

### Official Review · Reviewer_b1cj · 2025-10-29

**Soundness:** 2
**Presentation:** 2
**Contribution:** 2
**Rating:** 2
**Confidence:** 4

**Summary:**

The authors present a new multimodal RAG benchmark (though the reviewer argued that it's a long-context multimodal benchmark), especially targeting the wearable AI user cases. The benchmark analyses ablate how models perform on single-turn conversation image search, image-web unified search, and the multi-turn conversational scenarios. The results on this benchmark shows that large open-source and proprietary models still struggle with hallucination, especially when given low-quality images.

**Strengths:**

1. The synthetic benchmark dataset covers more conversation dynamics than prior benchmarks for factual question answering as shown in Figure 1 (See more in the weakness). Also, it's good that most images from this benchmarks are collected from real-human using egocentric wearable headset.
2. It's interesting that the paper targets wearable AI use cases with low-quality images. This setting is different from a lot of other relevant benchmarks.

**Weaknesses:**

1. Benchmark Dataset Design
	- The reviewer gets confused if it's a RAG benchmark, a search-augmented benchmark, or a long-context benchmark? Based on the description in Section 2 and Section 4.1, It seems that the authors provide an API function (tool) for these VLM to use and always assume the model would use it. If that's the case, it's more like a long-context QA benchmarks cuz the search part is fixed now. If not, it can be a search-augmented where the models might be able to decide whether they want to call the search. However, as the API is pre-determined, the paper seems to be far away from an RAG bnchmark studying the performance of a system including retrievers and generators as a whole.
	- The reviewers question if the benchmark is specifically targeting the wearable AI use cases or is developed to be "comprehensive" for multiple scenarios. The reviewer feels that the authors can make the benchmark at a clear position by picking either one bot not both. Beyond that, while images are manually collected, questions are semi-synthesized (task 1 and 2) or fully LLM generated (task 3). The reviewer would like to know more details for the dataset, especially the question distribution, intention, and examples. As a concrete suggestion, what's the difference between the curated dataset and the real human search queries in Search Arena where multi-turn questions are all from real human?

2. Experimental Design
	- The paper provides an image-based API and a text-search API search tool for models. However, as shown in Figure 3, using image-based search gets at most 58% recall, which limits models' performance at the beginning. Based on this biased, sub-optimal experimental design, the authors then draw several conclusions that seems to be questionable. For instance, for "entity recognition is much harder when relying solely on visual information compared to leveraging text clues" (L419-L420), the reviewer then want to know if it's because the choice of ViT-L/14@336px too weak. If the recall of image-search is the same or similar as text(web) retrieval, the reviewer would then convince the claim.
	- The reviewer wants to know more about the prompt to these models and the definition of truthness, missing, and hallucination. For prompting, it matters a lot as the authors now allow the model to answer "I don't know". Also, the reviewer is not sure if the incorrect in Line 315 identical as Hallucination in the remaining table. Finally, it's really confusing that the truthfulness can be negative in Tables, which the reviewer wants to know how it is computed. For multi-round conversations, do authors ensure the search models not to get the same content or not?

Overall, it's great that we have a new collected images, but the reviewer wants to know more details about the position/design of this benchmark. Also, the reviewer believes that the initial experiments/analyses on this benchmark data can be further strenthened.

**Questions:**

Please read the weakness section.

---

> ### Author Response · Authors · 2025-11-22
> **Clarifying Benchmark Dataset Design: CRAG-MM is a Flexible RAG Benchmark**
>
> We thank the reviewer for the constructive feedbacks and thoughtful questions!
>
> We clarify that **CRAG-MM is designed as a RAG benchmark**, but one that supports **multiple evaluation modes** depending on the user’s goals. The benchmark provides (1) question–answer pairs, (2) a retrieval corpus, and (3) optional search APIs. These components allow researchers to evaluate both retrieval and generation, together or separately.
>
> **First, the provided search APIs are optional tools intended to help users rapidly prototype MM-RAG systems.** This mode is particularly useful for challenge participants who need a ready-to-use retrieval interface. Even in this setting, the system is not reduced to a long-context QA task: models still need to decide **how to construct search queries, whether and how to crop images, and which information to extract**, in addition to **generating grounded answers** based on retrieved content.
>
> **Second, users can build and evaluate their own retrievers using the benchmark’s retrieval corpus.** This directly supports the study of RAG systems as a whole, including the retrieval component. By providing a shared corpus, CRAG-MM also enables **fair comparison**, eliminating discrepancies caused by unequal access to proprietary search engines or changing indices.
>
> **Third, the benchmark can evaluate built-in MM-RAG systems whose retrieval modules are not exposed to the user.**   In these cases, CRAG-MM assesses the system’s full retrieval–generation pipeline in a black-box manner. This is how we evaluate SOTA MM-RAG systems in the paper (see Section 5.2).
>
> Overall, CRAG-MM is explicitly designed to support MM-RAG evaluation, while also remaining flexible for different usage scenarios. The search APIs provide convenience not constraints, and do not fix the retrieval step. The benchmark therefore is a **RAG benchmark**.

---

> ### Author Response · Authors · 2025-11-22
> **Clarifying Benchmark Dataset Design: Position and Differentiation**
>
> Re: "The reviewers question if the benchmark is specifically targeting the wearable AI use cases or is developed to be "comprehensive" for multiple scenarios. The reviewer feels that the authors can make the benchmark at a clear position by picking either one bot not both."
>
> CRAG-MM is designed **primarily for wearable AI applications use cases**, and we will clarify this positioning in the paper. The benchmark is “comprehensive” in the sense that it captures the **diversity and variability naturally present in wearable scenarios**, including five types of image-quality issues, six question types, varying entity popularity, different information dynamism levels, and multi-turn interactions. These dimensions reflect real-world wearable challenges rather than an attempt to cover all possible scenarios. With that being said, we point out that the benchmark contain non-egocentric images, and many questions are meaningful even in non-wearables scenarios (e.g., "who made this book into a movie"), so can be used more broadly.
>
> &nbsp;
>
> Re: "Beyond that, while images are manually collected, questions are semi-synthesized (task 1 and 2) or fully LLM generated (task 3). The reviewer would like to know more details for the dataset, especially the question distribution, intention, and examples. As a concrete suggestion, what's the difference between the curated dataset and the real human search queries in Search Arena where multi-turn questions are all from real human?"
>
> We clarify that all single-turn (Tasks 1 and 2) and multi-turn (Task 3) questions were created through a **human-in-the-loop process**, not purely synthesized. While LLMs were used to draft candidate questions and answers for multi-turn conversation, **human annotators validated and revised every conversation**. For each sample, annotators ensured (1) the naturalness of the query, (2) appropriate difficulty, and (3) factual correctness of the answer. They removed or rewrote queries that are too simple or unnatural, corrected factual errors in the answers, and annotated all associated metadata.  Our preliminary analysis showed that this hybrid approach yields greater question diversity than fully human-crafted data alone.
>
> The design dimensions for CRAG-MM, domains, image-quality issues, question types, were inspired from **real user interactions with a commercial wearable device**, ensuring the dataset reflects authentic wearable-AI usage patterns. This leads to several clear differences from Search Arena:
>
> (1) **Modality**: CRAG-MM requires visual grounding (e.g., “When was this book published?”), whereas Search Arena evaluates text-only queries.
>
> (2) **Intent**: Search Arena focuses on open-domain knowledge questions; wearable users instead ask **environment-dependent, visually anchored queries**, such as “What brand is this?”.
>
> (3) **Answer Style**: CRAG-MM answers are usually concise and entity-focused, whereas Search Arena often involves long-form, explanatory responses.
>
> Please see the full domain and question-type distributions in Table 3, Sec 3.2, and examples in Figure 5 in the Appendix.

---

> ### Author Response · Authors · 2025-11-22
> **Addressing Experimental Design:  Robustness of Conclusions and Search API Flexibility**
>
> Re: "The paper provides an image-based API and a text-search API search tool for models. However, as shown in Figure 3, using image-based search gets at most 58% recall, which limits models' performance at the beginning. Based on this biased, sub-optimal experimental design, the authors then draw several conclusions that seems to be questionable."
>
> We agree that performance of RAG systems can be largely bounded by the recall of its retrievers. However, we believe the conclusions we draw in the paper are not compromised by the recall of the provided image-search API, for the following reasons.
>
> 1. **Our conclusion about the difficulty of entity recognition is not based on our image-search API.** The statement “entity recognition is much harder when relying solely on visual information compared to leveraging text clues” is derived from benchmarking **SOTA commercial MM-RAG systems with their own built-in retrieval engines.** In this evaluation, we rely only on our **questions and answers**, not on the provided search APIs. These industry systems use proprietary, highly optimized retrievers, likely far stronger than our ViT-L/14@336px setup, yet they still struggle significantly when no textual clues are provided. This indicates that the difficulty is **inherent to the wearable visual input**, not a consequence of our provided retriever.
>
> 2. **The lower recall of the benchmark’s image-search API reflects real-world wearable challenges, not a biased experimental design.** Wearable images naturally suffer from occlusion, blur, motion, poor lighting, and long-tail entities with limited public imagery (e.g., Walmart-exclusive apparel). These challenges are intrinsic to the problem. High recall on such images is extremely difficult because the underlying visual signal is often weak and noisy. Future **users can nevertheless build and evaluate their own retrievers using the benchmark’s retrieval corpus**. By providing a shared corpus, CRAG-MM also enables **fair comparison**, eliminating discrepancies caused by unequal access to proprietary search engines or changing indices.
>
> 3. **Low baseline scores are not solely attributed to image search failures.** To examine whether low image-search recall alone drives the difficulty, we re-evaluated models by **excluding the subset of questions where the entity was not retrieved by the image-search API**. As shown in the table below, accuracy remains only **56% and 61% on single- and multi-turn respectively**, and truthfulness only **39% and 42%**, highlighting that the task remains challenging beyond retrieval. Recognizing the entity is only the first step; constructing effective search queries and generating correct, grounded answers from web results remain difficult, validating the usefulness of CRAG-MM.
>
> In summary, the benchmark’s conclusions are not driven by a weak image retriever. They reflect fundamental challenges of wearable imagery and multimodal retrieval-augmented generation, as validated by both strong commercial systems and controlled subset analyses. We will clarify these points in the revised manuscript.
>
> &nbsp;
>
> | | | Model | Acc. | Miss. | Hallu. | Truth. | Early Stop. |
> | :--- | :--- | :--- | ---: | ---: | ---: | ---: | ---: |
> | **Single-turn** | **MM-LLM** | Llama 3.2 90B | 30.3 | 37.4 | 32.3 | -2.1 | - |
> | | | Gemini 2.5 Flash | 39.3 | *36.5* | 24.2 | 15.0 | - |
> | | | GPT-5 Mini | *39.7* | 42.5 | *17.9* | *21.8* | - |
> | | **Task 1** | Llama 3.2 90B | 16.8 | 67.1 | 16.1 | 0.7 | - |
> | | | Gemini 2.5 Flash | 42.2 | *34.6* | 23.2 | 19.0 | - |
> | | | GPT-5 Mini | *43.5* | 40.4 | *16.0* | *27.5* | - |
> | | **Task 2** | Llama 3.2 90B | 34.3 | 46.0 | 19.7 | 14.5 | - |
> | | | Gemini 2.5 Flash | **55.5** | **20.1** | 24.4 | 31.0 | - |
> | | | GPT-5 Mini | 55.1 | 29.1 | **15.8** | **39.3** | - |
> | **Multi-turn** | **MM-LLM** | Llama 3.2 90B | 42.4 | *24.8* | 32.8 | 12.9 | 64.2 |
> | | | Gemini 2.5 Flash | 28.1 | 59.0 | **12.9** | 16.2 | 87.9 |
> | | | GPT-5 Mini | *47.4* | 36.3 | 16.3 | *29.8* | *62.8* |
> | | **Task 3** | Llama 3.2 90B | 38.8 | 44.1 | 17.0 | 20.4 | 71.7 |
> | | | Gemini 2.5 Flash | 55.4 | 24.1 | 20.6 | 32.8 | 54.5 |
> | | | GPT-5 Mini | **60.9** | **22.9** | *16.2* | **42.4** | **44.2** |
>
> Table 1. Performance of straightforward MM-RAG solutions on a subset of 65% single-turn QAs and 85% multi-turn sessions in CRAG-MM, where retrieval obtains correct facilitating information. Even when the answer exists in the retrieval results, the truthfulness is up to only 39.3\% and 42.4% for single- and multi-turn QAs respectively.

---

> ### Author Response · Authors · 2025-11-23
> **Addressing Experimental Design:  Clarifying Evaluation Metrics, Prompting, and Multi-turn Retrieval Logic**
>
> Re: "The reviewer wants to know more about the prompt to these models and the definition of truthness, missing, and hallucination. For prompting, it matters a lot as the authors now allow the model to answer "I don't know". Also, the reviewer is not sure if the incorrect in Line 315 identical as Hallucination in the remaining table. Finally, it's really confusing that the truthfulness can be negative in Tables, which the reviewer wants to know how it is computed."
>
>
> We apologize for the confusions and will revise the paper to clarify. We answer the questions briefly below.
>
> **Truthfulness, Missing, and Hallucination.**
>
> We adopt the Truthfulness score defined in [1], where each model response is categorized into **accurate**, **missing**, or **incorrect** (which we refer to as **hallucination** throughout the paper). And the **Truthfulness** score equals to
>
> Accuracy (percentage of accurate answers) - Hallucination (percentage of incorrect answers).
>
> A score can therefore be **negative** when the model produces more incorrect answers than correct ones, reflecting severe hallucination.
> We will revise the paper to use “hallucination” consistently instead of mixing “hallucinated” or “incorrect.”
>
> **Prompting and Allowing Abstention.**
>
> The prompts explicitly allow the model to answer “I don’t know” to encourage abstention rather than hallucination, consistent with the design of the Truthfulness metric. We included the prompt in Appendix A5.2.
>
> Reference:
>
> [1] https://arxiv.org/abs/2406.04744
>
> &nbsp;
>
> Re: "For multi-round conversations, do authors ensure the search models not to get the same content or not?""
>
> **Context and Retrieval Content for Multi-turn Conversation.**
>
> To ensure we fully address the reviewer’s question, we provide an explanation below regarding how retrieval content is managed in multi-turn settings. We interpret the question as inquiring whether we explicitly prevent the retrieval of duplicate content across different turns.
>
> - **Query-Dependent Retrieval.** In our baseline multi-turn RAG implementation, retrieved content is dynamic because the search queries evolve. For each turn, we employ Llama-3.2-11B-Instruct model to generate a rewritten search query based on the current question, prior conversation history, and retrieval results from image search. As the conversation focus shifts, the rewritten queries change, naturally leading to different retrieved content.
>
> - **Deduplication Policy.** Within turn, our search API is designed to ensure that a single API call does not return duplicate web pages. Across turns, we do not filter out pages in current turn simply because they appeared in previous turns. We allow overlap because a follow-up question might require re-examining the same source document for different details (e.g., Turn 1 asks for the author, Turn 2 asks for the publication date from the same wiki page). In practice, due to the query rewriting described above, the retrieved results typically vary across turns.
>
> - **Conversation History.** Previously generated responses from the same VLM model under evaluation were preserved, ensuring the model has full context of the dialogue flow.
>
> *If this explanation does not fully capture the reviewer’s concern regarding "same content," we would be grateful for a specific clarification and are happy to provide further details.*

---

> ### Author Response · Authors · 2025-11-23
> **Request for Clarification on Strengthening Experimental Analysis**
>
> Re: "the reviewer believes that the initial experiments/analyses on this benchmark data can be further strengthened."
>
> We thank the reviewer for the valuable feedbacks. If the reviewer has suggestions regarding how the experiments can be further strengthened beyond the discussion, we would like to incorporate.

---

> > ### Comment · Reviewer_b1cj · 2025-11-23
> > **Reviewer's Response**
> >
> > Thanks for the detailed rebuttal.
> >
> > For the position, I now get the point that the word "comprehensive" is within the field of wearable AI. While I appreciate the authors want to enable this data for evaluating both RAG / search-augmented generation systems, I feel that the current writing is still confused in terms of the **evaluation pipeline and protocol: how each module is called and what's the variable the authors want to study and what are fixed**. Perhaps I need to read the updated version as the authors promised earlier. Also, I do feel that it's important to compare the proposed benchmark dataset with other prior work cuz the question seems to be short and unnatural as shown in Figure 5.
> >
> > For the justification of "Robustness of Conclusions and Search API Flexibility", while the reviewer understand the difficulty of doing an open-source pipeline, the ViT-L/14@336px seems to be weak in 2025... and okay back to the first point, I still did not fully understand how ViT-L/14@336 works with Gemini or GPT as their input would be the raw image rather than the ViT embedding or what's the role of these two models in the above rebuttal experimental results?
> >
> > For the metric, I appreciate the authors to clarify the source. While I won't say the current definition is wrong, the hallucination/incorrect is still a bit confusing when first reading it (as mentioned by other reviewers as well).
> >
> > Overall, I'm excited to see the updated PDF and appreciate the authors' detailed response but at the same time, I feel that the paper might require significant editing to clarify all these points.

---

> > > ### Author Response · Authors · 2025-12-02
> > > **Response to follow-up questions from Reviewer b1cj**
> > >
> > > We thank the reviewer for continued engagement. We acknowledge the need for greater clarity regarding the evaluation protocol and have revised the manuscript to address this, including Sec 3.3 (Retrieval Contents), Sec 5 (Experiment Design and Result) and Sec 4 (Metrics). We provide specific clarifications below.
> > >
> > > 1. **Clarification on Evaluation Pipeline & Protocol.**
> > >
> > > In our straightforward solution evaluation (Mode 1), the MM-LLMs receive the **question**, **raw input image**, the same **constructed retrieval content** (image metadata + web page snippets) from the fixed Search APIs, and **conversation history** (when applicable). This setup isolates the model’s ability to generate grounded answers from realistic, noisy retrieval content.
> > >
> > > In our industry SOTA evaluation (Mode 3), only the questions, raw input images and ground truth are fixed. These systems are evaluated end-to-end in a full black-box manner, with variable RAG pipelines.
> > >
> > > 2. **The Role of ViT-L/14.**
> > >
> > > ViT-L/14@336px is strictly a retrieval encoder. The outputs of image search are provided as text metadata for downstream tasks, such as query rewriting or question-answering. The encoder is never seen by the MM-LLMs (e.g. Gemini/GPT), nor the embeddings. It **only serves to** power the **image search API** in evaluating the straightforward solutions.
> > >
> > > **Why use ViT-L/14?**
> > > We chose ViT-L/14 to ensure the retrieval pipeline is fully **open-source, reproducible and non-proprietary**. While we agree it is not current SOTA, our results (Sec 5.2) show that even industry SOTA systems with proprietary, highly-optimized retrievers struggle on this benchmark. This confirms that the difficulty stems from the **wearable QA with egocentric images**, not merely the choice of our open-source encoder.
> > >
> > > 3. **Question Naturalness and Comparison with Prior Work.**
> > >
> > > The reviewer notes that questions appear short and unnatural. We clarify that **CRAG-MM questions are intentionally short**, reflecting realistic user behavior when interacting with wearable devices. Unlike web search or web-based chat, wearable interactions, which are primarily voice-driven, are typically short [1-3]. For example, a user walking through a grocery store would rarely ask “Please read this product label and tell me if it contains gluten”; they would instead ask, “Is this gluten-free?” CRAG-MM is designed to represent such daily, practical interactions with wearable devices. Quantitively, CRAG-MM has an average query length of 9 words, consistent with other wearable VQA or general purpose VQA benchmarks, such as WearVQA [4] (\~10 words) and MM-Vet [7] (\~11 words).
> > >
> > > On the other hand, we believe **CRAG-MM questions are natural**. To ensure naturalness and realism, question design drew directly from observations of actual wearable device interactions. We defined domains, question types, and image quality challenges that mirror real usage; trained annotators with detailed guidelines emphasizing the creation of **plausible, user-initiated queries**; and performed multi-stage reviews to filter out trivial, unnatural, or template-like questions. Furthermore, an automated audit using GPT-4o confirmed that **over 99%** of the CRAG-MM queries are natural and coherent user requests for Wearable scenarios.
> > >
> > > Finally, we explicitly **compared CRAG-MM with existing factual QA benchmarks**. As detailed in **Table 1**, CRAG-MM is differentiated from existing factual QA benchmarks by its focus on egocentric perspective, retrieval corpus/API, multi-turn dynamics, ensuring it addresses a unique and realistic gap in the field.
> > >
> > > We will add additional representative examples in the next revision to better convey the benchmark’s characteristics and enhance clarity for readers.
> > >
> > >
> > > 4. **Evaluation Metrics.**
> > > We follow previous works including [5] to separate correct/missing/incorrect, and [6] to compute truthfulness. We have revised the paper to consolidate hallucination/incorrect to prevent confusion.
> > >
> > >
> > > References:
> > >
> > > [1] Gurari, Danna, et al. "Vizwiz grand challenge: Answering visual questions from blind people." Proceedings of the IEEE conference on computer vision and pattern recognition. 2018.
> > >
> > > [2] Song, Inpyo, et al. "Video question answering for people with visual impairments using an egocentric 360-degree camera." arXiv preprint arXiv:2405.19794 (2024).
> > >
> > > [3] Muralidharan, Deepak, et al. "Noise robust named entity understanding for voice assistants." NAACL. 2021.
> > >
> > > [4] Chang, Eun, et al. "WearVQA: A Visual Question Answering Benchmark for Wearables in Egocentric Authentic Real-world scenarios".
> > >
> > > [5] Sun, Kai, et al. "Head-to-tail: How knowledgeable are large language models (LLMs)? AKA will LLMs replace knowledge graphs?." 2024.
> > >
> > > [6] Yang, Xiao, et al. "Crag-comprehensive rag benchmark." Advances in Neural Information Processing Systems 37 (2024): 10470-10490.
> > >
> > > [7] Yu, Weihao, et al. "Mm-vet: Evaluating large multimodal models for integrated capabilities". arXiv
> > > preprint arXiv:2308.02490, 2023.

---

### Official Review · Reviewer_WBeU · 2025-11-01

**Soundness:** 2
**Presentation:** 3
**Contribution:** 2
**Rating:** 2
**Confidence:** 3

**Summary:**

This paper introduces a new benchmark dataset CRAG-MM, that is leveraged to evaluate multi-modal RAG systems specifically when applied to wearable scenarios. The benchmark is designed based on a diverse set of question styles and practical tasks. The data for the retrieval have been designed to reflect the imperfections associated with real-world situations. Specifically, the corpus used for textual sources of information included significant amount of noise (irrelevant passages and URLs) and images with varying quality levels, the view point, unideal cropping, lighting levels, etc. The benchmark dataset had been shown to be challenging uncovering shortcomings associated with existing MM-RAG systems.

**Strengths:**

The paper propose an interesting benchmark for multi-modal RAG specifically useful for situations pertaining to wearables, which are becoming more and more common place. This benchmark is extremely relevant for the current and future developments of the field. Moreover, the design of the question types, tasks and the data quality feeding the RAG systems, makes the benchmark realistic, hence making the benchmark useful and reflective of the true performance of the systems evaluated with it in real world scenarios. The design choices behind the question types, tasks and data are well motivated and discussed. The paper overall is well-written and presented.

**Weaknesses:**

While the benchmark is well motivated and the rationale behind the dataset design and data types is clearly presented, important details are missing regarding how the dataset was actually created and how the quality of its questions was ensured. I encourage the authors to consult works such as MMQA and SMMQG, which provide comprehensive documentation of their data collection and validation processes, including the use of crowd-sourced annotators, inter-annotator agreement checks, and bias mitigation strategies. The absence of comparable methodological transparency in this paper raises concerns about the reliability of the dataset and, consequently, the validity of the reported evaluation results. This represents a fundamental weakness that should be addressed to strengthen the paper’s overall contribution. Similar issues/shortcomings also persist as related to the auto-evaluator evaluation.

Moreover, since all evaluated RAG systems rely exclusively on the retrievers provided by the benchmark, their performance is inherently bounded by the recall and quality of those retrievers. The paper itself reports relatively low retrieval recall which sets an upper limit on achievable QA accuracy. As a result, it becomes difficult to disentangle whether the observed limitations stem from the benchmark’s retrieval pipeline or from the reasoning capabilities of the tested models.

**Questions:**

Along the lines of the major weaknesses above, the following questions at the bare minimum need to be addressed

- Section 3.2 mentions that annotators “created plausible questions for wearable devices” and “recorded the ground truth answers” but does not describe annotation instructions, inter-annotator agreement, validation stages, or any bias mitigation strategy.

- Similarly, the multi-turn QA generation (in Section 3.2) relies heavily on prompting Llama-3.2-90B and then “review by annotators,” yet provides no quantitative metrics for human verification.

- The paper’s “Reproducibility Statement” (page 10) claims detailed description of dataset creation, but Appendix A.2 and A.3 still omit annotation guidelines, quality-control sampling, or reviewer calibration.

- Provide more details on the auto-evaluator evaluations

- The limitations of the benchmark performance based on the performance limitations of the retrievers need to be addressed and further discussed.

---

> ### Author Response · Authors · 2025-11-22
> **Addressing Methodological Transparency: Details on Annotation, Validation, and Bias Mitigation**
>
> We agree that Section 3.2 provides only a brief description to conserve space. We detailed concrete steps for constructing the QA sets from the web and from the KG in Appendix A.2. Following your suggestion, we plan to add more detailed explanations of our human QA-construction pipeline in Appendix A.2, including 5 stages: guideline design, annotator qualification, QA and metadata curation, verification and correction, and expert audit. We next repeat them here and will upload our revision soon.
>
> **Stage 1: Guideline Design.** Annotators were tasked with identifying fine-grained entities (e.g., "Golden Retriever" rather than "Dog") within images and generating question-answer pairs and associated metadata such as image-query category (e.g. plant, local) and question type. We enforced strict constraints to ensure validity.
> - Visual grounding: To prevent models from relying on language priors [1,2], annotators were required to use anaphoric pronouns (e.g., "Where is this building?"), rendering the query textually ambiguous and forcing the model to rely on visual context to resolve the reference.
> - Stratified query type distribution: to prevent the benchmark from skewing towards either trivial recognition or purely textual trivia, we enforced a ~50/50 split between simple (recognition / knowledge) and complex queries as defined in Sec 2.1. This addresses answer-distribution bias (as discussed in OK-VQA [3]).
> - Factual determinism: questions were restricted to those with indisputable, deterministic answers. Subjective (e.g., "Is this cute?") or open-ended questions were prohibited to ensure consistent evaluation.
> - Safety & privacy: Face blur was properly applied prior to annotation. Images containing PII or integrity violations will be rejected.
> - Other constraints: such as to avoid repeating queries, avoid asking questions that’s too general such as “what is this”, and avoid asking questions with yes/no answers.
> These constraints are designed to comply with our benchmark principles and mitigate common bias in VQA dataset.
>
> **Stage 2: Annotator Qualification.** Candidates need to review the guidelines, pass a knowledge quiz, and complete a training queue with 100 QA creation jobs. Qualification required passing a high-quality threshold evaluated against a fine-grained error rubric.
>
> **Stage 3: QA and Metadata Curation.** Qualified annotators proceeded to the production phase to construct image-question-answer triples. For each image, annotators were tasked with: (1) identifying valid fine-grained entities; (2) formulating a visual-anaphoric question, e.g. "when was this building built?"; and (3) providing a deterministic answer backed by a verified source URL and associated metadata (e.g., entity source, query dynamism).
>
> **Stage 4: Verification and Correction.** Following initial curation, a separate pool of high-quality annotators conducted a full-pass verification, mitigating self-confirmation bias - where annotators overlook ambiguities in their own writing. Annotators focused on correcting metadata errors, fixing grammatical issues and rejecting samples with data validity or safety issues.
> - This stage resulted in the modification of ~25% of samples and the rejection of ~2% of samples due to irresolvable quality or safety violations.
>
> **Stage 5: Expert Audit.** A final quality audit was conducted by three in-house experts on a random subset. They assessed the quality of the triples and the correctness of the associated metadata, achieving an inter-annotator agreement rate of ~90%.
>
> References:
>
> [1] https://arxiv.org/abs/1612.00837
>
> [2] https://arxiv.org/abs/1712.00377
>
> [3] https://arxiv.org/abs/1906.00067

---

> ### Author Response · Authors · 2025-11-22
> **Addressing Methodological Transparency: Quantitative Metrics for Multi-turn QA**
>
> All multi-turn conversations generated by the Llama-3.2-90B model were **validated and, when necessary, revised by human annotators**. For each conversation, annotators reviewed the naturalness and difficulty of the questions as well as the factual correctness of the answers. They rewrote queries that were overly simple or unnatural, corrected any factual errors in the answers, and annotated all associated metadata. In the final dataset, **79% of conversations contain at least one human-edited query or answer**, with an average of **4.8 turns modified** per conversation. We will add these details to the revision.

---

> ### Author Response · Authors · 2025-11-22
> **Addressing Methodological Transparency: Details on Auto-evaluator Evaluations**
>
> We took several steps to ensure that the auto-evaluator produces accurate and reliable assessment:
>
> 1. To reduce evaluation ambiguity, we curated and validated **high-quality ground-truth answers** so that auto-evaluation is reduced to a semantic-matching task. Our process included three quality-control stages:
>
>     (1) **Qualification and annotation.** All annotators received training and were required to pass a qualification test aligned with our grading guidelines. We will include the detailed guidelines in the revised manuscript.
>
>     (2) **Initial data creation.** During QA construction, one annotator verified the factual correctness of each answer and made correction if needed.
>
>     (3) **Review and expert audit.** A second annotator conducted a full independent review, followed by a sampled audit by an expert annotator. Approximately 7% of answers were corrected during this stage, resulting in a final ground-truth correctness rate of **~95%**.
>
> 2. We conducted extensive validation of the LLM-judge. **The GPT-4o–based judge achieves 99% accuracy and 91% F1** against the manual judge labels (Table 7 in Appendix A4.1), demonstrating strong reliability. To ensure the reliability of the manual judge labels used for validating the LLM-judge, we conducted the following:
>
>     (1) **Data collection.** Randomly sampled diverse responses from both straightforward MM-RAG solutions and public competition submissions across all tasks (12k/7k/5k samples for Tasks 1–3), aiming to reduce sampling bias.
>
>     (2) **Qualification and annotation.** Annotators passed two qualification steps (one for the task objective, the other for the grading guideline) and followed strict guidelines for quality grading to evaluate the model response quality against the ground truth, focusing on factual correctness, relevance and contextual understanding. Detailed guidelines will be included in the revised manuscript.
>
>     (3) **Expert audit.** In-house linguists audited a random sample of the annotations, achieving **~92%** inter-annotation agreement and confirming **88% / 91%** quality for single-turn and multi-turn response evaluation.
>
>
> We will add these details in the revised manuscript.

---

> ### Author Response · Authors · 2025-11-22
> **Addressing Retrieval Recall Limitation: Oracle Analysis and Search API Flexibility**
>
> Re: "The limitations of the benchmark performance based on the performance limitations of the retrievers need to be addressed and further discussed."
>
> &nbsp;
>
> We thank the reviewer for the thoughtful questions!
>
> We agree that performance of a RAG system is largely bounded by the recall of its retrievers. However, this does not fundamentally limit the value of CRAG-MM for three reasons.
>
> 1. **The provided search APIs are convenience tools, not constraints.** They are intended to help users and challenge participants rapidly build MM-RAG systems. Importantly, CRAG-MM includes a full retrieval corpus, enabling users to implement and evaluate **their own retrievers**. As shown in Figure 3a, an oracle retriever can reach **93.4% recall** on the provided image set.
>
> 2. **Low scores of baseline methods are not solely due to image search failures.** We removed the subset of questions where the entity was not retrieved by the image search API (see table below). Even with correct entity recall, accuracy remains only **56% and 61% on single- and multi-turn respectively**, and truthfulness only **39% and 42%**, highlighting that the task remains challenging beyond image retrieval and validating the usefulness of CRAG-MM.
>
> 3. **Low recall reflects real-world challenges for wearable AI, not benchmark limitations.** Wearable images are often unfocused, occluded, poorly lit, or depict long-tail entities with low presence in public sources (e.g., a specific Walmart-exclusive clothing item). These conditions mirror the real challenges faced in commercial wearable AI systems. Notably, even SOTA MM-RAG systems, which have strong built-in search and vision capabilities based on their own retrieval corpus, exhibit low truthfulness on CRAG-MM (Table 5), underscoring the gap in current MM-RAG technologies for wearable scenarios.
>
> &nbsp;
>
> | | | Model | Acc. | Miss. | Hallu. | Truth. | Early Stop. |
> | :--- | :--- | :--- | ---: | ---: | ---: | ---: | ---: |
> | **Single-turn** | **MM-LLM** | Llama 3.2 90B | 30.3 | 37.4 | 32.3 | -2.1 | - |
> | | | Gemini 2.5 Flash | 39.3 | *36.5* | 24.2 | 15.0 | - |
> | | | GPT-5 Mini | *39.7* | 42.5 | *17.9* | *21.8* | - |
> | | **Task 1** | Llama 3.2 90B | 16.8 | 67.1 | 16.1 | 0.7 | - |
> | | | Gemini 2.5 Flash | 42.2 | *34.6* | 23.2 | 19.0 | - |
> | | | GPT-5 Mini | *43.5* | 40.4 | *16.0* | *27.5* | - |
> | | **Task 2** | Llama 3.2 90B | 34.3 | 46.0 | 19.7 | 14.5 | - |
> | | | Gemini 2.5 Flash | **55.5** | **20.1** | 24.4 | 31.0 | - |
> | | | GPT-5 Mini | 55.1 | 29.1 | **15.8** | **39.3** | - |
> | **Multi-turn** | **MM-LLM** | Llama 3.2 90B | 42.4 | *24.8* | 32.8 | 12.9 | 64.2 |
> | | | Gemini 2.5 Flash | 28.1 | 59.0 | **12.9** | 16.2 | 87.9 |
> | | | GPT-5 Mini | *47.4* | 36.3 | 16.3 | *29.8* | *62.8* |
> | | **Task 3** | Llama 3.2 90B | 38.8 | 44.1 | 17.0 | 20.4 | 71.7 |
> | | | Gemini 2.5 Flash | 55.4 | 24.1 | 20.6 | 32.8 | 54.5 |
> | | | GPT-5 Mini | **60.9** | **22.9** | *16.2* | **42.4** | **44.2** |

---

### Author Response · Authors · 2025-11-23

We thank the reviewers for the thoughtful questions and valuable feedbacks! We are revising the paper and will upload a new version next week.

---

### Author Response · Authors · 2025-11-26
**Revision 1 Uploaded: Summary of Major Updates**

We have uploaded a revised manuscript incorporating the reviewers' constructive feedback. All edits are **highlighted in blue**, except minor wording change. Key updates include:

- **Formalized Evaluation Modes (Sec 3.3)**: We clarified the usage of retrieval content and search APIs, and added Section 3.3.3 to explicitly define the three evaluation modes supported by CRAG-MM: (1) API-based retrieval & summarization, (2) Customized retrieval & summarization and (3) Black-box end-to-end evaluation.

- **Experimental Design & Analysis (Sec 5)**:
     - Sec 5.1: Added detailed protocols for straightforward RAG baselines, leaderboard-winner solutions, and industry SOTA systems.
     - Sec 5.2.1: Added a new ablation study evaluating MM-LLMs on the subset of queries with successful image search coverage (full results in Appendix A.6.2). This isolates model performance from low retrieval recall.
     - Sec 5.2.2 and 5.2.3: Refined the results discussion to better articulate system comparisons and caveats.

- **Metric Definitions (Sec 4)**: Revised to provide precise definitions of the Truthfulness metric for both single-turn and multi-turn QAs.

- **Data Rigor (Appendix A.2)**: Significantly expanded the documentation on annotation guidelines, quality control pipelines, and privacy/consent protocols.

- **Auto-evaluator Validation (Appendix A.5)**: Added detailed evaluation process for the auto-evaluator.

- **Formatting**: Reformatted all prompt templates in the Appendix for improved readability.

---

### Meta-Review · Area_Chair_rCBR · 2026-01-05

**Summary:**

This paper proposes a benchmark for evaluating multimodal RAG systems in the space of wearable AI. The paper addresses a clear gap and has a strong relevance. It was found that the dataset reflects challenging real-world scenarios, and provides good empirical evaluation. The reviewers raised several concerns, including those pertaining to insufficient transparency  in the methodology, affecting reliability,, limitations in the retrieval pipeline that could confound analysis, confusion with metrics etc.

**Reviewer Concerns:**

Although the authors have addressed the reviewer feedback in their rebuttal, some concerns may remain (e.g. choices such as ViT-L/14@336px being weak in 2025).

**Reviewer Scores:**

The current scores not only reflect the concerns/questions the reviewers had, but also the extent of contribution this paper makes. Given that, and the extent of the questions and rebuttal, I would conjecture the average review would stay the same if there was a chance for a full interaction.

---

### Decision · Program_Chairs · 2026-01-26

Reject